# Dominance modifiers at the Arabidopsis self-incompatibility locus retain proto-miRNA features and act through non-canonical pathways

Rita A. Batista[1,¤], Eléonore Durand[1], Monika Mörchen[1], Jacinthe Azevedo-Favory[2], Samson Simon[1], Manu Dubin[1], Vinod Kumar[3], Eléanore Lacoste[4], Corinne Cruaud[5], Christelle Blassiau[1], Matteo Barois[1], Anne-Catherine Holl[1], Chloé Ponitzki[1], Nathalie Faure[1], William Marande[6], Sonia Vautrin[6], Isabelle Fobis-Loisy[7], Jean-Marc Aury[4], Sylvain Legrand[1], Ute Krämer[3], Thierry Lagrange[2], Xavier Vekemans[1], Vincent Castric[1]*

**1** Univ. Lille, CNRS, UMR 8198 – Evo-Eco-Paleo, Lille, France, **2** CNRS, Laboratoire Génome et Développement des Plantes, UMR 5096, Perpignan, France, and Université Perpignan Via Domitia, Laboratoire Génome et Développement des Plantes, UMR 5096, Perpignan, France, **3** Department of Molecular Genetics and Physiology of Plants, Faculty of Biology and Biotechnology, Ruhr University Bochum, Bochum, Germany, **4** Génomique Métabolique, Genoscope, Institut François Jacob, CEA, CNRS, Univ Evry, Université Paris-Saclay, Evry, France, **5** Genoscope, Institut François Jacob, CEA, Université Paris-Saclay, Evry, France, **6** INRAE - CNRGV French Plant Genomic Resource Center, Castanet Tolosan, France, **7** Laboratoire Reproduction et Développement des Plantes, Université de Lyon, ENS de Lyon, UCB Lyon 1, CNRS, INRAE, INRIA, Lyon, France

¤ Current address: Max Planck Institute for Biology, Department of Algal Development and Evolution, Tübingen, Germany
* vincent.castric@univ-lille.fr

## Abstract

Self-incompatibility in flowering plants is a common mechanism that prevents self-fertilization and promotes outcrossing. In Brassicaceae, the self-incompatibility locus is highly diverse, with many alleles arranged in a complex dominance hierarchy and exhibiting monoallelic expression in heterozygote individuals. Monoallelic expression of the pollen self-incompatibility gene is achieved through the action of sRNA precursors that resemble miRNAs, although the underlying molecular mechanisms remain elusive. Here, we engineered *Arabidopsis thaliana* lines expressing components of the *Arabidopsis halleri* self-incompatibility system, and used a reverse genetics approach to pinpoint the pathways underlying the function of these sRNA precursors. We showed that they trigger a robust decrease in transcript abundance of the recessive self-incompatibility genes, but not through the canonical transcriptional or post-transcriptional gene silencing pathways. Furthermore, we observed that single sRNA precursors are typically processed into hundreds of sRNA molecules with a variety of sizes, abundance levels and ARGONAUTE loading preferences. Our results suggest that these seemingly arbitrary processing characteristics are essential for establishing the self-incompatibility dominance hierarchy, as they enable a single sRNA precursor from a dominant allele to effectively repress multiple recessive

**Data availability statement:** The AGO-IP and sRNA sequencing data are deposited in the Gene Expression Omnibus (GEO) database under the SubSeries references GSE249507 and GSE249508. The Illumina and Oxford Nanopore sequencing data of the five A. halleri individuals carrying alleles Ah65, Ah60, Ah03, Ah33, and Ah19 are available in the European Nucleotide Archive (ENA) under the project number PRJEB70880. The RNA-seq data used to infer the SCR01 transcription start site are available at ENA-EBI under the accession numbers ERR14765081, ERR14765082, and ERR14765083. The S-locus sequences used in this study are a combination of previously published data [15,19] and data generated in this study. All allele sequences can be found in GenBank under the following references: Ah01 – this study, PP396894; Ah02 – this study, PP396895; Ah03 – this study, PP537897; Ah04 – KJ461484; Ah10 – KM592810 and KM592817; Ah12 – KJ772373 and KJ772377; Ah13 – KJ461479 and KJ461483; Ah19 – this study, PP537898; Ah20 – FO203486; Ah25 – this study, PP396896; Ah28 – KJ461478; Ah29 – KM592803; Ah33 – this study, PP537899; Ah60 – this study, PP537900; Ah65 – this study, PP537901.

**Funding:** R.A.B. received funding from the European Molecular Biology Organization (EMBO; https://www.embo.org) through a Postdoctoral Fellowship. V.C. received funding from the European Research Council (ERC; https://erc.europa.eu) (Grant #648321) and from the French National Research Agency (Agence Nationale de la Recherche; https://anr.fr) (Grant ANR-18-CE02-0020-01). T.L. received funding from the French National Research Agency (Agence Nationale de la Recherche; https://anr.fr) through the LABEX TULIP (Grant ANR-10-LABX-41) and the EUR TULIP-GS (Grant ANR-18-EURE-0019). U.K. received funding from the Deutsche Forschungsgemeinschaft (DFG; https://www.dfg.de) (Grants Kr1967/10-1, Kr1967/10-2) and the European Research Council (ERC; https://erc.europa.eu) (Grant 788380). The funders had no role in the study design, data collection and analysis, decision to publish, or preparation of the manuscript. The molecular biology platform of the Lille group is supported by the Région Hauts-de-France and the Ministère de l'Enseignement Supérieur et de la Recherche (CPER Climibio and CPER Ecrin grants), and the European Fund for Regional Economic Development.

**Competing interests:** The authors have declared that no competing interests exist.

alleles, thus providing a unique example of how small RNAs mediate gene silencing within a highly complex regulatory network.

## Author summary

Many plants avoid self-fertilization through a system of self-incompatibility, which allows them to recognize and reject their own pollen. In plants like those in the Arabidopsis family, self-incompatibility operates through a "lock-and-key" system: the female part of the flower is the lock, and pollen carries the key. To successfully fertilize different plants, pollen must express only one type of key, while silencing other versions present in its genome. This silencing, known as dominance, is controlled by small RNA molecules produced from small RNA pre-cursors, similar to microRNAs. However, how small RNAs achieve this selective silencing has remained unclear. To address this, we engineered *Arabidopsis thaliana* - a species that does not naturally have self-incompatibility - to carry genes from a related self-incompatible species and studied the small RNAs they produce. We found that each precursor generates hundreds of small RNAs with different sizes and properties, which are collectively important to silence "unwanted" key variants present in pollen, and surprisingly, these RNAs do not follow the typical silencing pathways. Our work provides a new insight into how plants control the expression of the lock-and-key components required for self-incompatibility, a system that ensures successful reproduction and the maintenance of genetic diversity.

## Introduction

Self-incompatibility is a widespread self-recognition mechanism in hermaphroditic flowering plants that prevents selfing and enforces outcrossing. In Brassicaceae, self-pollen recognition is controlled at the sporophytic level by a single non-recombining self-incompatibility locus (*S*-locus) containing two protein-coding genes: the female component encoding a trans-membrane receptor (SRK) expressed in papillae cells, and the male component consisting of an SCR peptide (referred to as SP11 in the Brassica nomenclature) produced in the anther tapetum and deposited on the surface of the pollen grains. The SRK and SCR proteins function as a receptor-ligand system, such that pollen germination is stopped when a pair of cognate SRK and SCR proteins interact upon pollen deposition on the pistil. This process relies on a cascade of signaling events involving additional, non-*S*-locus proteins, and ultimately prevents self-fertilization [1–5]. The *S*-locus has been subject to an intense long-term balancing selection, which favors the maintenance of a high number of distinct *S*-alleles [6–9]. Although heterozygosity is typically very high at the *S*-locus, monoallelic expression of the male *SCR* gene is the rule, following a transitive and mostly linear dominance hierarchy among *S*-alleles [10–12]. These

dominance interactions increase reproductive success at the level of the diploid individual by ensuring that only one of the two *SCR* alleles is expressed in pollen. As a result, pollen is rejected only by pistils expressing the same allelic specificity, rather than by pistils carrying either of the two specificities present in the heterozygote male, thereby increasing the number of compatible mating partners [13]. In *Arabidopsis lyrata* and *A. halleri*, respectively, 56 and 43 *S*-alleles have been documented so far [7,14], which are divided into four main dominance classes - class IV being the most dominant and class I the most recessive [15,16].

Underlying these dominance interactions are dominance modifiers - genetic elements dedicated to altering dominance relationships between other alleles in *trans* [17]. In the particular context of the self-incompatibility system these dominance modifiers are located in the *S*-locus and take the form of sRNA-generating loci that are hypothesized to modulate the expression of the *SCR* gene [18]. Similarly to miRNAs genes, these loci are transcribed into an RNA molecule that is predicted to fold and form a hairpin structure, which is subsequently cleaved into sRNAs. In *A. halleri*, eight distinct families of sRNA precursors, defined based on sequence similarity, are predicted to regulate dominance interactions both between and within the four *S*-allele dominance classes [19]. Interestingly, in Brassica where only two dominance classes exist, only two sRNA precursor families have been identified [12,20–22], suggesting that there is an association between the complexity of the dominance network and the total number of dominance modifiers. The two sRNA precursors of Brassica (*Smi* and *Smi2*) have each been suggested to produce a 24-nt sRNA with sequence similarity to the 5' region of *SCR.* The presence of this sRNA is associated with decreased expression of the recessive *SCR* allele, as well as the deposition of DNA methylation within and around the targeted region in tapetum cells [20–22]. DNA methylation at the targeted allele is established during pollen development and reaches maximal levels at the stage when *SCR* is normally expressed, consistent with a role in transcriptional repression. This has led to the hypothesis this 24-nt sRNA mediates *de novo* DNA methylation at the *SCR* locus, although the underlying molecular mechanisms remain unclear. This is reminiscent of what is observed in several plant species where 21/24-nt sRNAs derived from miRNAs genes or inverted repeats can be loaded onto proteins of the ARGONAUTE4 (AGO4) clade and subsequently elicit *de novo* DNA methylation at target loci, such as transposable elements (TEs) or protein-coding genes, through the RNA-directed DNA Methylation (RdDM) pathway [23–27]. *De novo* DNA methylation mediated by the RdDM pathway could represent a general mechanism for *SCR* repression in the Brassicaceae self-incompatibility system. However, DNA methylation at recessive *SCR* loci has so far only been demonstrated for *Smi* and *Smi2* [20–22], and the mechanisms underlying its deposition remain unexplored.

In Brassica, the dominance hierarchy of *S*-alleles was proposed to rely on the combinatorial effect of single nucleotide variants at the *Smi* and *Smi2* sequences and at their respective *SCR* target sites [21]. In Arabidopsis, where more dominance classes exist, the dominance network is hypothesized to rely on two properties: i) recessive alleles carry more target sites and these sites are more generalist (*i.e.,* they are targeted by more sRNA precursors families than dominant alleles); and ii) sRNA precursors of dominant alleles are more generalist (*i.e.,* they target a higher number of alleles than those sRNA derived from precursors in more recessive alleles) [19]. Despite this observation, the molecular features allowing sRNA precursors to have a broad targeting spectrum remain to be identified. In addition, among the eight families of sRNA precursors previously predicted to control the dominance hierarchy in Arabidopsis, only one has been subject to formal experimental validation [19].

In this study, we aimed at identifying the molecular pathway underlying the sRNA-mediated dominance interactions between *S*-alleles in Arabidopsis. To achieve this, we engineered and characterized a series of *A. thaliana* lines recapitulating the self-incompatibility phenotype of *A. halleri*, and used these lines to show that two different sRNA precursors from dominant *A. halleri* *S*-loci trigger a decrease in abundance of recessive *SCR* transcripts. We thus provide direct proof that this heterologous approach can be used to validate sRNA precursor function, both at the phenotypic and at the molecular level. This system further allowed us to investigate the molecular basis of *SCR* transcript repression by *S*-locus sRNA precursors. Because the precursors we investigated target non-coding regions of *SCR*, and since *Smi* and *Smi2* have been previously shown to trigger *de novo* DNA methylation at SCR target sites in Brassica [20–22], we tested whether key

components of the canonical RdDM pathway are required for *S*-locus sRNA precursor function in Arabidopsis. We specifically examined the roles of the ARGONAUTE proteins AGO4 and AGO6, as well as the POLIV and POLV subunits of the DNA-dependent RNA polymerase. Our analyses indicate that none of these components are required for the activity of the sRNA precursors, suggesting that they act independently of the canonical RdDM pathway.

Additionally, we reveal that despite their structural similarity to miRNAs, *S*-locus-derived sRNA precursors produce a large population of sRNAs with different sizes, sequences, ARGONAUTE loading preferences and target sites. We show that this molecular heterogeneity is important to maximize the number of targeted *S*-alleles, thus allowing one dominant allele to silence multiple recessive alleles. Overall, our results show how the distinctive molecular features of *S*-locus sRNAs contribute to effective gene silencing within a complex regulatory network, shedding light on their role in shaping dominance interactions in the context of an important reproductive phenotype.

## Results

### The sRNA precursors Ah04mir1887 and Ah20mirS3 control dominance interactions at the *S*-locus by decreasing the transcript level of the recessive *SCR01* allele

To study in detail the molecular mechanisms controlling the activity of dominance modifiers in the self-incompatibility system of *A. halleri*, we first recreated the self-incompatibility reaction of *A. halleri* in the selfing species *A. thaliana*, since the latter is a more tractable study model (Fig 1). To do this, we expressed the *SRK01* receptor and its cognate *SCR01* ligand in pistil and pollen donors, respectively [19,28,29]. Manual crosses between these plants revealed that the germination of pollen from a *SCR01*-expressing male was abolished when deposited on pistils of an *SRK01*-expressing female, the hallmark of a self-incompatible reaction (Figs 1C and 1D, WT panels, and S7). Given the challenges faced in prior studies to reconstruct this reaction in the Col-0 ecotype, all *A. thaliana* plants used in this study are of the C24 ecotype, where the self-incompatibility can be faithfully recreated as shown both in this, and in other studies [28,30,31].

Among the eight known families of *A. halleri* sRNA precursor loci, only one - mirS3 - has been experimentally validated to regulate dominance between two distinct *S*-alleles: specifically, the dominance of allele Ah20 over Ah01, mediated by Ah20mirS3 sRNAs, which target the *SCR01* intronic region (Fig 1B) [19]. In this study, we set out to further investigate the molecular properties of Ah20mirS3, as well as examine if the yet uncharacterized mir1887 family could also function as a dominance modifier. For this, we focused on testing the role of the sRNA precursor Ah04mir1887 in controlling the dominance of allele Ah04 over Ah01, whose sRNAs are predicted to target the promoter region of *SCR01*, immediately upstream of the TSS (Figs 1A and S1). We thus generated transgenic lines expressing Ah04mir1887, introgressed this transgene into an *SCR01*-expressing male, and used these plants to pollinate *SRK01*-expressing females, testing whether the presence of the Ah04mir1887 sRNA precursor locus could transform the previously incompatible interaction between *SCR01* and *SRK01* into a compatible one (Fig 1C). In the particular case of Ah20mirS3, the transgenic line previously generated by Durand et al. [19] and used here expresses both Ah20mirS3 and Ah20mir1887 because of their close physical proximity within the Ah20 allele (Fig 1B). To specifically isolate the impact of Ah20mirS3 in the dominance interaction of Ah20 over Ah01, we developed the *SCR01*\**mir1887* transgenic line, which contains five point mutations designed to fully disrupt the slight homology observed between some Ah20mir1887 sRNAs and *SCR01* (Figs 1B and S2). We then used this modified *SCR01*\**mir1887* line for all subsequent tests on the functional role of Ah20mirS3.

We observed that combining either Ah04mir1887 (Interaction A), or Ah20mirS3 (Interaction B) with *SCR01* results in a switch from an incompatible to a compatible pollen germination phenotype, which is accompanied by a dramatic decrease in the abundance of the *SCR01* transcript, as measured by RT-qPCR (Fig 1C and 1D, WT panels). This shows that both Ah04mir1887 and Ah20mirS3 act as dominance modifiers in their respective allelic interactions, and that they accomplish this by decreasing *SCR01* transcript abundance, thus preventing SRK-SCR cognate recognition and allowing pollen germination to occur. Moreover, these results demonstrate that *SCR* transcript abundance can be modulated by multiple sRNA precursor families that use distinct target sites in the *SCR* locus.

 

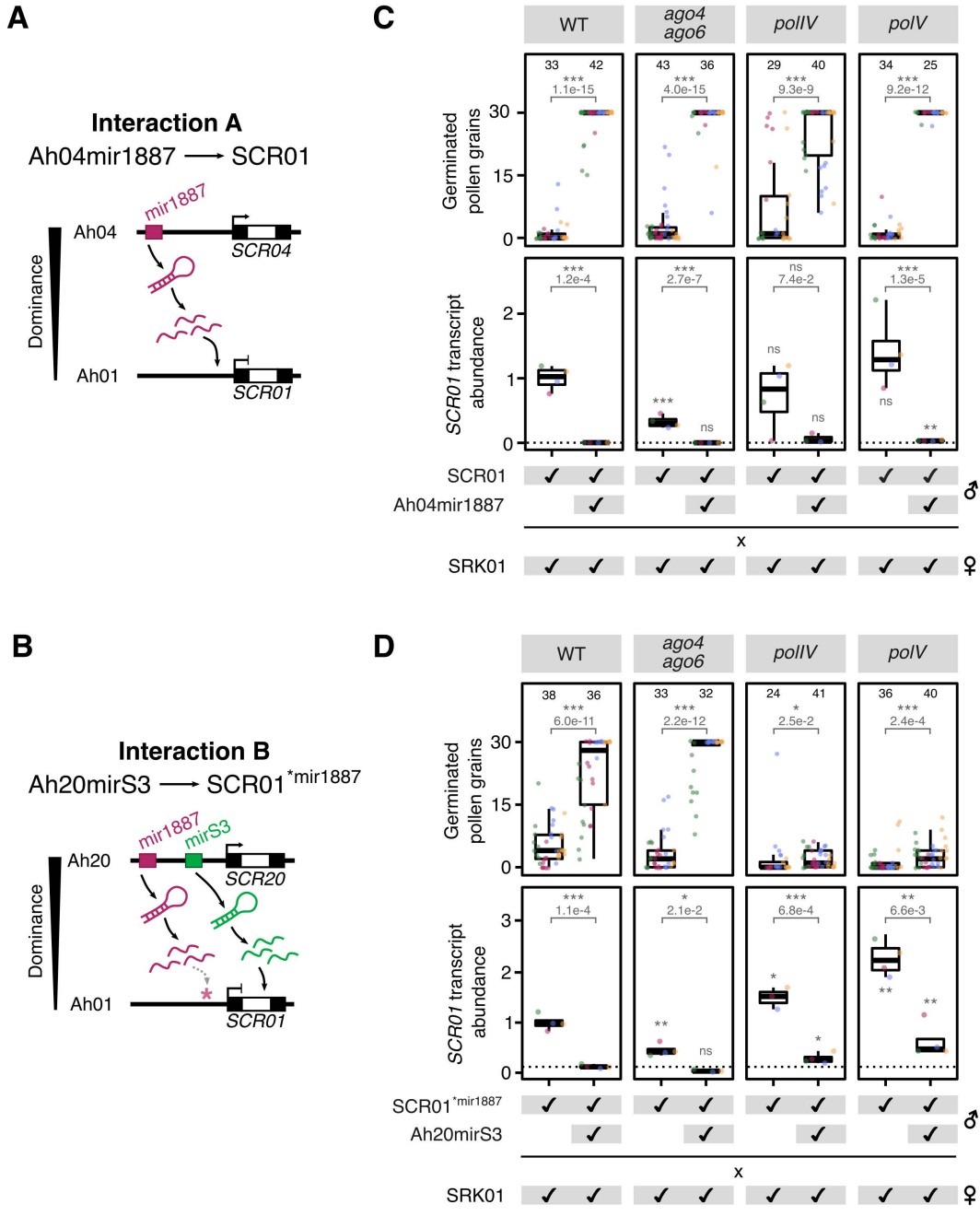

**Fig 1. The RdDM pathway is not required for the activity of Ah04mir1887 and Ah20mirS3. (A,B)** Dominance interactions studied in this paper. The black and white regions within *SCR* genes correspond to exons and introns, respectively. **(A)** Interaction A focuses on the role of the sRNA precursor Ah04mir1887 in mediating the dominance of Ah04 over Ah01. **(B)** Interaction B focuses on the role of Ah20mirS3 in mediating the dominance of Ah20 over Ah01. *S*-locus sRNA precursors consist of inverted repeats that form hairpins which are processed into multiple sRNAs showing sequence homology to specific *SCR* regions: Ah04mir1887 sRNAs show homology to the 5' region of *SCR01*, while Ah20mirS3 sRNAs show homology to the intronic region (white) of *SCR01*. Because sRNAs produced from the Ah20mir1887 precursor show homology to the wild-type (WT) *SCR01* 5' region, and to study the specific effect of Ah20mirS3 in Interaction B, the *SCR01*\*mir1887 line containing five point mutations in the *SCR01* 5' region was created, which disrupts the homology with Ah20mir1887 sRNAs. **(C,D)** Phenotypic and expression assay evaluating the role of key RdDM components on the action of Ah04mir1887 **(C)**, and Ah20mirS3 **(D)**. Crosses between *SRK01*-expressing pistils and *SCR01*-expressing pollen donors were performed in WT and RdDM mutant backgrounds in the absence or presence of the respective sRNA precursor. The number of germinated pollen grains of the indicated genotype was counted on SRK01 pistils. *SCR01* expression in immature buds of the indicated genotype was measured by RT-qPCR and normalized to

the WT level. Four to five biological replicates (represented by the differently colored dots) were analyzed for pollen germination and *SCR01* expression. Multiple dots of the same color in the pollen germination panel represent technical replicates (different pistils pollinated by the same pollen donor). Dots of the same color across the pollen germination and the *SCR01* expression panels correspond to the same pollen donor, allowing direct comparison between phenotype and expression. Numbers on top of columns indicate the total number of siliques sampled for pollen germination assays. Pollen germination differences were assessed using a two-sided Wilcoxon rank-sum test comparing the presence vs. absence of sRNA precursor in each mutant background. Differences in *SCR01* expression were assessed using log2-transformed normalized expression levels and unpaired two-tailed Welch's t-tests. Pairwise comparisons between genotypes are indicated by brackets, and comparisons between each mutant background and the corresponding WT control (in the presence or absence of sRNA precursors) are shown above each boxplot. Statistical results for all pairwise comparisons are detailed in S3 Table.

## Decrease in *SCR01* transcript abundance does not depend on RdDM

Next, we aimed at elucidating the exact molecular mechanism responsible for the reduction of *SCR01* transcript abundance triggered by the two *S*-locus sRNA precursors. Given their resemblance to miRNAs genes, one potential scenario is that sRNAs produced by Ah04mir1887 and Ah20mirS3 could trigger post-transcriptional gene silencing (PTGS) of the *SCR01* transcript. However, since these sRNAs are predicted to target the promoter and intronic region of *SCR01*, and since the substrate of PTGS is generally thought to be the mature mRNA [32], this does not support PTGS as the pathway responsible for the decrease in *SCR01* transcript abundance. An alternative scenario is that Ah04mir1887 and Ah20mirS3 sRNAs could trigger the RdDM pathway, resulting in the deposition of DNA methylation in their respective *SCR01* target sites, and reduced *SCR01* transcription. Interestingly, previous studies have established that the silencing of recessive alleles in the homologous self-incompatibility system of *Brassica* relies on sRNA-mediated *de novo* DNA methylation deposition at *SCR* target loci, although the specific pathway underlying this process remains elusive [12,20,21]. Thus, we focused our attention on the RdDM pathway, and set out to test if mutating key components of this pathway would prevent Ah04mir1887 and Ah20mirS3 from reducing *SCR01* transcript abundance. To circumvent the absence of RdDM mutant alleles in the C24 ecotype, we generated and validated C24 *ago4 ago6* CRISPR/Cas9 mutant alleles (S3 Fig), and used the previously published C24 *polIV* and *polV* knock-out alleles [33].

To our surprise, none of the RdDM mutants impaired the activity of the tested sRNA precursors. In the case of Ah04mir1887 (Interaction A), *SCR01* transcript abundance was reduced in all mutant backgrounds to a similar level as in wild type, which was reflected by a compatible pollen germination phenotype (Fig 1C). Similarly, the presence of Ah20mirS3 (Interaction B) was associated with a decrease in *SCR01* transcript in all mutant backgrounds, showing that this sRNA precursor is still functional when RdDM is impaired (Fig 1D). And while in the *ago4 ago6* mutant background the reduction in *SCR01* transcript led to a compatible pollen phenotype, in both *polIV* and *polV* mutant backgrounds the reduction in *SCR01* transcript level was significant but insufficient to elicit a compatible pollen phenotype (Fig 1D). This is likely explained by the fact that in Interaction B the basal *SCR01* transcript level is increased in both *polIV* and *polV* mutant backgrounds when compared with wild type (Fig 1D, *polIV* and *polV* panels vs. WT panel), pointing to an effect of these mutants on the expression of the *SCR01* transgene. This suggests that the *SCR01*\*mir1887 construct might be intrinsically targeted for repression by RdDM, a phenomenon often observed in transgenic lines [34] (note that the *SCR01* and *SCR01*\*mir1887 transgenes used in Interaction A and B, respectively, correspond to two independent transgenic lines with different insertion sites; see Methods section). Indeed, our data shows that a substantial reduction of *SCR01* transcript abundance is required for a compatible pollen germination phenotype, since even a minimal transcript abundance can trigger an incompatibility reaction (S4 Fig).

To determine whether these Arabidopsis sRNA precursors, like in Brassica [12,20,21], affect DNA methylation at *SCR01* target sites, we performed BiSulfite Amplicon Sequencing (BSAS) [26,35]. *SCR01* is expressed specifically in tapetum cells of immature floral buds, which are difficult to isolate manually in *A. thaliana* due to their small size and embedment within other floral tissues. To circumvent this, we used BSAS, since this technique relies on PCR amplification of bisulfite-converted DNA to achieve high coverage and detect subtle methylation changes. We sampled 8–10 biological

replicates of *SCR01* transgenic lines in the presence or absence of Ah04mir1887 and Ah20mirS3 (S5 Fig). In parallel, pollen from the same plants was used to pollinate *SRK01*-expression pistils to confirm the phenotypic self-incompatibility response. In this way, DNA methylation (S5C Fig) and pollen germination (S5B Fig) measurements are paired, allowing a direct comparison between the *SCR01* methylation level and the phenotype for each individual.

BSAS yielded an average coverage of ~1400× per cytosine across *SCR01* target sites and the 800 bp flanking regions (S5 Table). These regions showed high cytosine methylation (75–95% across all sequence contexts), similar to what is typically observed in TEs rather than genic sequences (S5 Fig). Despite this, no significant changes were detected in the presence of Ah04mir1887 or Ah20mirS3, suggesting that the decrease in *SCR01* transcript abundance is not accompanied by changes in DNA methylation at the target sites (S5 Fig). Stratifying BSAS reads by fraction of methylated cytosines per read revealed a subset of unmethylated reads, which we hypothesize may originate from tapetum cells where *SCR01* is expressed (S6 Fig). However, the abundance of these reads was similarly unaffected by the sRNA precursors.

While the interpretation of this BSAS data is limited by the use of bulk floral tissue and the unexpectedly high baseline DNA methylation, these results are in line with our phenotypic assays (Fig 1), which indicate that canonical RdDM is unlikely to be responsible for the reduction in *SCR01* transcript abundance triggered by *S*-locus sRNA precursors. Despite this apparent independence from RdDM, both Ah04mir1887 and Ah20mirS3 strongly and robustly decrease *SCR01* transcript levels, which we show is required to abolish the self-incompatibility reaction.

## Ah04mir1887 and Ah20mirS3 hairpins undergo heterogeneous processing to generate populations of diverse sRNAs

*S*-locus sRNA precursors, including those of the mir1887 and mirS3 families studied here, consist of inverted repeats that are transcribed and predicted to form a RNA hairpin structure containing a 5' arm, a terminal loop and a complementary 3' arm (Fig 2A) [19], akin to the structures formed by precursor miRNA (pri-miRNA) molecules. To delve deeper into the molecular properties of Ah04mir1887 and Ah20mirS3, and to explore potential non-canonical pathways through which these precursors may act, we conducted several sRNA sequencing experiments on both transgenic *A. thaliana* and wild type *A. halleri* plants harboring Ah04mir1887 and Ah20mirS3. The resulting data allowed us to compile a comprehensive database encompassing all sRNAs molecules sequenced for each precursor, irrespective of their species origin. Using these data, we observed that dicing of *S*-locus sRNA precursor hairpins leads to numerous sRNAs – at least 106 in the case of Ah04mir1887, and 224 in the case of Ah20mirS3 (Fig 2B). These sRNAs range from 18 to 25 nt in length, with no single length clearly overrepresented (S8A and S8B Fig), and their abundances vary by several orders of magnitude (Figs 2B, S8C and S8D).

Given the large population of sRNAs produced by a single precursor, we aimed at evaluating the functional relevance of each sRNA in controlling the abundance of the *SCR01* transcript. For this, we used a previously published algorithm [19], which attributes a targeting score based on the homology between a single sRNA and the *SCR* genomic region. A previous study demonstrated that in multiple *A. halleri S*-locus heterozygotes, a targeting score equal to, or above 18 is associated with phenotypic dominance and a sharp decrease of the recessive *SCR* transcript level [10]. Using this algorithm, we found that 12 out of the total 106 sRNAs produced by Ah04mir1887, and 33 out of the total 224 sRNAs produced by Ah20mirS3, have a base-pairing score of ≥ 18 and are thus predicted to target *SCR01*. This subset of putatively functional sRNAs similarly displays a variety of sizes and expression levels (Fig 2B, see grey box).

To further assess whether these molecular features are reproducible between the native *A. halleri* context and the transgenic *A. thaliana* context, we performed the same analyses as above, but using only the sRNAs detected in *A. halleri* samples (S9 Fig). Although fewer sRNAs were identified in *A. halleri*, their size distribution, abundances, and targeting spectrum closely mirror those observed in the combined sRNA dataset from both species (S9A–S9F Fig). We hypothesize that the reduced number of sRNAs detected in *A. halleri* is primarily due to the lower sequencing effort in this species compared to *A. thaliana* (S9G Fig), suggesting that the total number of sRNAs identified for each precursor is highly

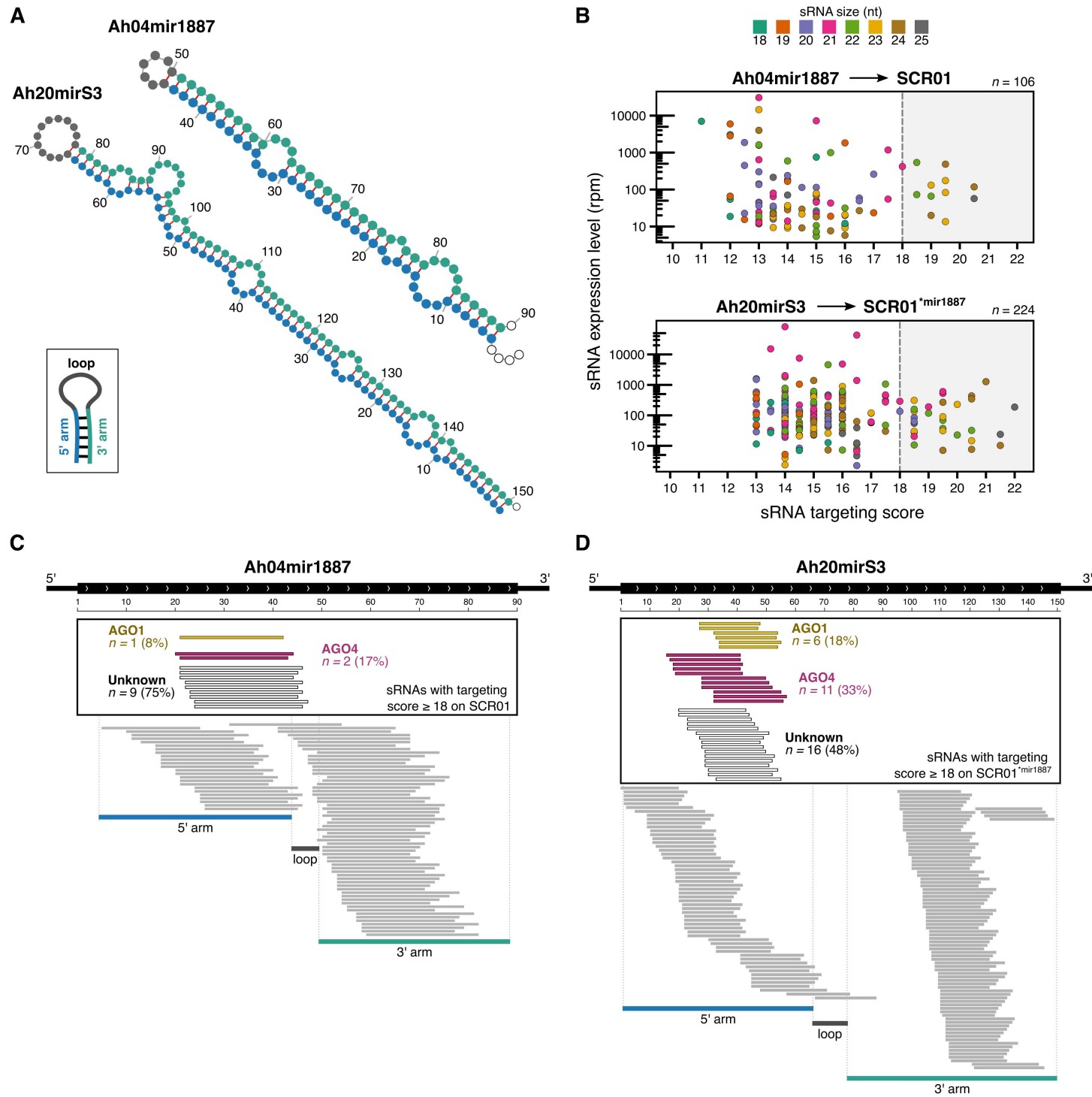

**Fig 2. Ah04mir1887 and Ah20mirS3 hairpins are processed into numerous sRNAs. (A)** Predicted RNA hairpin structures of the *S*-locus sRNA precursors Ah04mir1887 and Ah20mirS3. **(B)** Expression of Ah04mir1887 and Ah20mirS3 sRNAs as a function of their targeting score against *SCR01*. The dot color corresponds to the size of the sRNA. The grey box highlights sRNAs predicted to induce a reduction in *SCR01* transcript abundance (score ≥ 18). *n* corresponds to the total number of unique sRNAs identified for each sRNA precursor. **(C-D)** Genomic representation of the Ah04mir1887 and Ah20mirS3 loci, and their respective sRNAs. The black rectangle highlights sRNAs predicted to induce a reduction in *SCR01* transcript abundance (score ≥ 18). AGO loading information for each functional sRNA is indicated by the sRNA color (yellow – sRNA loaded predominantly in AGO1; magenta – sRNA loaded predominantly in AGO4; white – sRNAs with unknown loading profile), see Methods section *ARGONAUTE immunoprecipitation and sRNA sequencing* for more details.

dependent on sequencing depth, and that the saturation point was not reached in *A. halleri*. Overall, these data indicate that the transgenic *A. thaliana* system accurately recapitulates the features observed in the native *A. halleri* context, further supporting the use of *A. thaliana* as a more tractable system to study the molecular bases of self-incompatibility reactions. Based on these findings, we decided to proceed with the initially compiled sRNA database, which includes the sum of sRNAs sequenced in both *A. halleri* and *A. thaliana*, for further downstream analyses in this study.

To better understand the AGO loading preferences for each sRNA we performed AGO immunoprecipitation (IP) assays, followed by sRNA sequencing in the Ah04mir1887 and Ah20mirS3 *A. thaliana* transgenic lines. This showed that functional sRNAs (base-pairing score of ≥ 18) produced from either precursor are loaded both into AGO1 (characteristic of the PTGS pathway) and AGO4 (characteristic of the RdDM pathway) (Fig 2C and 2D). Notwithstanding this, a large part of these functional sRNAs (75% in Ah04mir1887 and 48% in Ah20mirS3) are not found to be loaded in either of these two AGO proteins, consistent with our observations that both the canonical PTGS and RdDM pathways are unlikely to underlie the activity of these sRNA precursors. We also noted that the majority of sRNAs loaded in AGO1 have a 5' terminal uracil, reflecting the known AGO1 preference for binding these sRNAs [36]. However, sRNAs loaded into AGO4, or with an unknown loading pattern, do not show a significant bias towards a specific 5' nucleotide (S10 Fig).

Despite the structural similarity between *S*-locus sRNA precursors and miRNAs, and the existence of shared features, such as the presence of mature 21-nt sRNAs and loading of some sRNAs into AGO1, it is evident that *S*-locus sRNA precursors deviate from the typical characteristics associated with canonical miRNAs, and cannot be characterized as such. Instead, our data show that Ah04mir1887 and Ah20mirS3 have an unusually staggered hairpin processing pattern with numerous sRNAs derived from both the 5' and 3' arms, which differ in size, abundance level and ability to associate with AGO proteins.

## Heterogeneous processing of *S*-locus sRNA precursors allows targeting of multiple *S*-alleles

We hypothesized that this heterogeneity in hairpin processing pattern could play an important role in the context of the complex *S*-locus dominance hierarchy by allowing a single sRNA precursor to exert dominance over multiple other alleles. To test this, we expanded our analysis of the function of Ah04mir1887 and Ah20mirS3 beyond the interaction with allele Ah01, and looked at their role within the *A. halleri* dominance network (Fig 3).

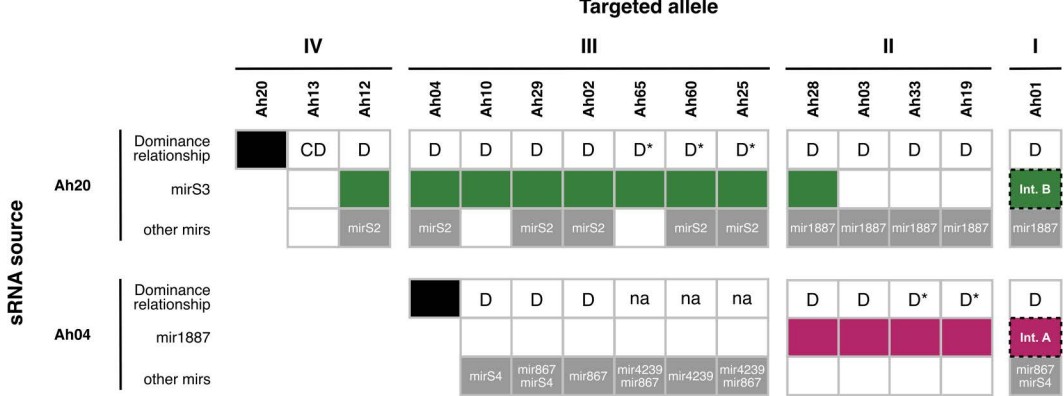

**Fig 3. Ah04mir1887 and Ah20mirS3 underlie numerous *S*-allele dominance interactions.** Schematic representation of the dominance relationship between alleles Ah20, Ah04 and other *S*-alleles in the current *A. halleri* dominance network. The observed dominance phenotypes are represented by D (dominant, as determined by phenotypic assays), CD (codominant, as determined by phenotypic assays), and na (no phenotypic data available). D* refers to a predicted dominance relationship, based on the difference between the dominance class of the sRNA source and the targeted allele [7,16]. The sRNA precursors underlying each dominance relationship are highlighted: green for mirS3, magenta for mir1887, and grey for other sRNA precursors. Interaction A (Ah20mirS3 → *SCR01*) and Interaction B (Ah04mir1887 → *SCR01*) are highlighted with dashed boxes to emphasize the focus of this study. The remaining sRNA precursors present in the Ah20 and Ah04 loci are included in the row "other mirs" to represent their potential contributions to dominance.

We compiled previously published phenotypic data obtained by controlled reciprocal crosses between different *A. halleri* S-locus heterozygotes to determine the position of allele Ah20 and Ah04 in the dominance hierarchy of a total of 15 S-alleles (Fig 3) [10,19,37]. From this, we inferred that Ah20 is dominant over 13 other S-alleles, with 10 of these interactions relying on the action of Ah20mirS3 (Fig 3), as determined by the sRNA-*SCR* targeting algorithm described above. Thus, Ah20mirS3 is not only able to mediate the repression of *SCR01*, but also the repression of at least nine other distinct *SCR* alleles. Similarly, Ah04mir1887 mediates the repression of at least five different *SCR* alleles that are recessive to Ah04 (Fig 3).

To test if the large population of sRNAs produced by these S-locus precursors is important in repressing multiple target alleles, we examined the role of each individual sRNA produced by both Ah04mir1887 and Ah20mirS3 (Fig 4): 11% of Ah04mir1887 sRNAs (Fig 4A, cluster 1), and 26% of Ah20mirS3 sRNAs (Fig 4B, cluster 1) are predicted to target at least one *SCR* allele within the inferred dominance network presented in Fig 3. As is the case in Interaction A and B, there does not seem to be a clear correlation between abundance and sRNA functionality in any of the analyzed interactions (Figs 2B and 4, see cluster 1 vs. cluster 2). Nevertheless, sRNAs that are predicted to be functional have a larger size (median of 23 nt vs. 21 nt), and in the case of Ah20mirS3, this is also associated with a larger number of sRNAs being loaded into AGO proteins (Fig 4, see cluster 1 vs. cluster 2). Additionally, according to our sRNA-*SCR* targeting algorithm, most S-alleles are redundantly targeted by multiple sRNAs (Fig 4).

When investigating in more detail the positional correspondence between sRNAs and the Ah04mir1887/Ah20mirS3 precursor hairpin structure we noticed that sRNAs are produced in similar amounts from both the 5' and the 3' arms of the hairpin (Fig 5). Each hairpin arm produces numerous sRNAs that show a staggered pattern, suggesting imprecise dicing and resulting in a population of sRNAs that show only a few nucleotide differences between them. In the case of Ah04mir1887, only the sRNAs produced from the 5' hairpin arm are predicted to be functional against the S-alleles tested here (Fig 5A). Remarkably, in the case of Ah20mirS3, each hairpin arm shows functional specialization since sRNAs from the 5' arm target all allele classes except Class IV, while sRNAs from the 3' arm exclusively target Class IV and Class III alleles (Fig 5B). Together, these observations suggest that the high number and heterogeneous processing of sRNAs is a conserved and important feature of S-locus sRNA precursors which allows targeting of multiple alleles and thus underlies their generalism.

## Discussion

### *S*-locus sRNA precursors act through non-canonical gene silencing pathways

Our characterization of self-incompatibility transgenic lines of *A. thaliana* contributes additional evidence to the work of Durand et al. [19]: collectively, these studies not only establish *A. thaliana* as a tractable system to study dominance interactions among *A. halleri* S-alleles, but also experimentally validate the role of two distinct sRNA precursor families in these interactions, even if the precise mechanisms underlying their action remain elusive. Our study reveals that some of the sRNAs generated in the processing of the S-locus sRNA precursors have a length of 24 nt and are loaded into AGO4, potentially indicating a role for RdDM. Nevertheless, *ago4 ago6*, *polIV* and *polV* mutations did not significantly affect the action of *A. halleri* S-locus sRNA precursors, and the presence of these precursors was not associated with DNA methylation changes in *SCR01*. Together these data suggest that canonical RdDM is not the primary mechanism through which Ah20mirS3 and Ah04mir1887 mediate the reduction in *SCR01* transcript levels.

Considering their hairpin structure and their ability to also generate AGO1-loaded 21-nt sRNAs, it could also be envisioned that these precursors could act through PTGS. Still, the location of target sites in the promoter and intronic regions of *SCR01* poses a conundrum, since PTGS is reported to occur in the cytoplasm, using mature mRNA as a cleavage substrate [32]. Notably, in animals, several lines of evidence point to miRNAs acting both in the cytoplasm and in the nucleus, with effects at the transcriptional and post-transcriptional levels [38–40]. In plants, it has been observed that an inverted repeat targeting the intron of the soybean gene *FAD12-1A* triggers cleavage of its precursor mRNA (pre-mRNA)

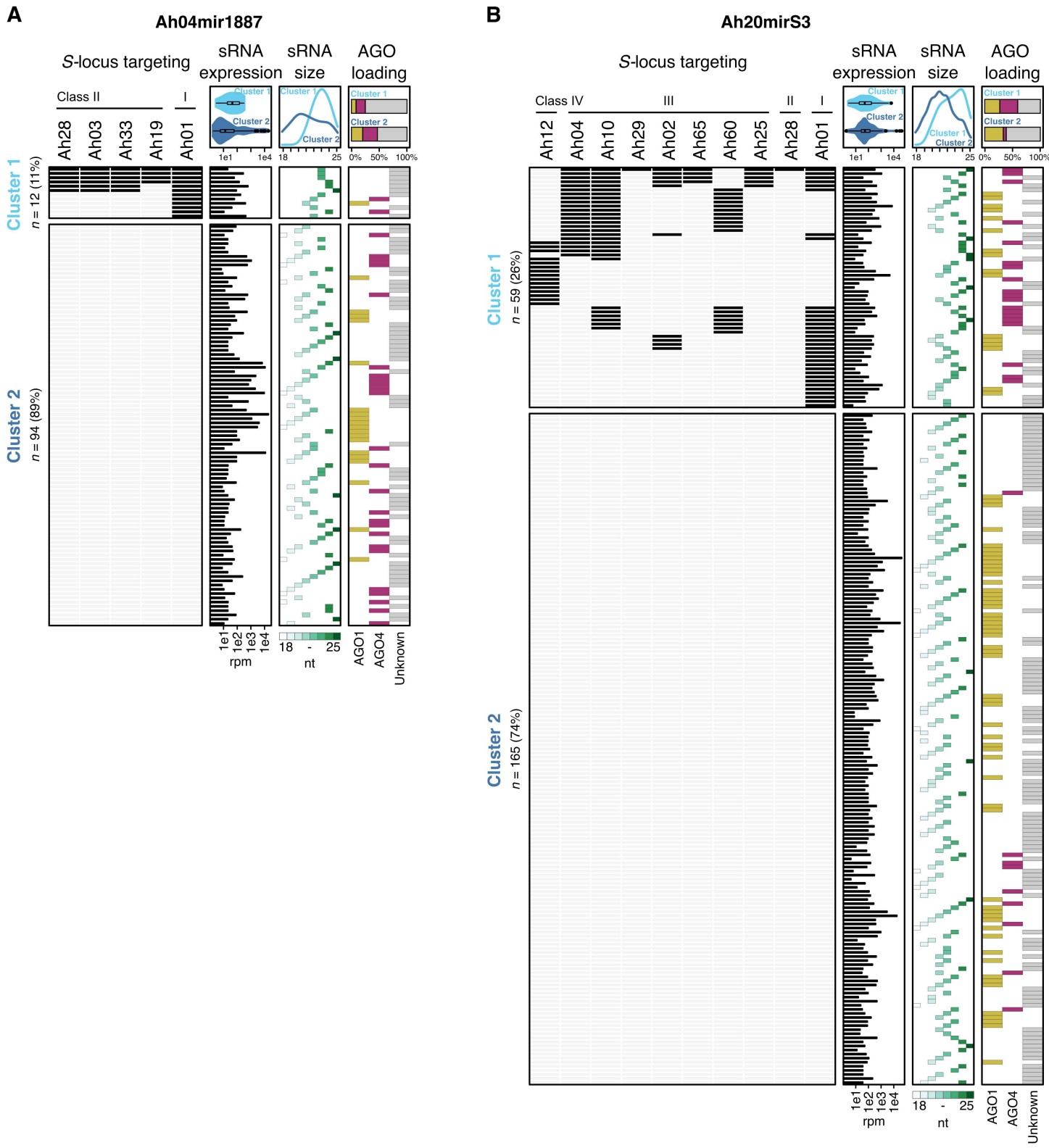

**Fig 4. Ah04mir1887 and Ah20mirS3 produce multiple sRNAs predicted to target one or more *S*-alleles.** Heatmaps showing the predicted interactions between individual sRNAs produced by Ah04mir1887 **(A)**, Ah20mirS3 **(B)** and the represented *S*-alleles. Each row corresponds to a single

sRNA and each column to an *S*-allele. Black rectangles indicate a predicted interaction with a targeting score ≥ 18. sRNAs are grouped into two clusters: Cluster 1, comprising sRNAs predicted to target at least one *S*-allele, and Cluster 2, comprising sRNAs with no predicted targets among the represented *S*-alleles. For each sRNA, it expression level, size, and AGO loading pattern are indicated alongside the heatmap. The top panels show summary plots comparing these features (expression, size, and AGO loading) between Cluster 1 (indicated in light blue) and Cluster 2 (indicated in dark blue). The AGO loading summary plot shows the cumulative percentage of sRNAs per cluster, loaded into AGO1 (yellow), AGO4 (magenta), or of unknown loading status (grey). *n* indicates the total number of distinct sRNAs produced by each precursor.

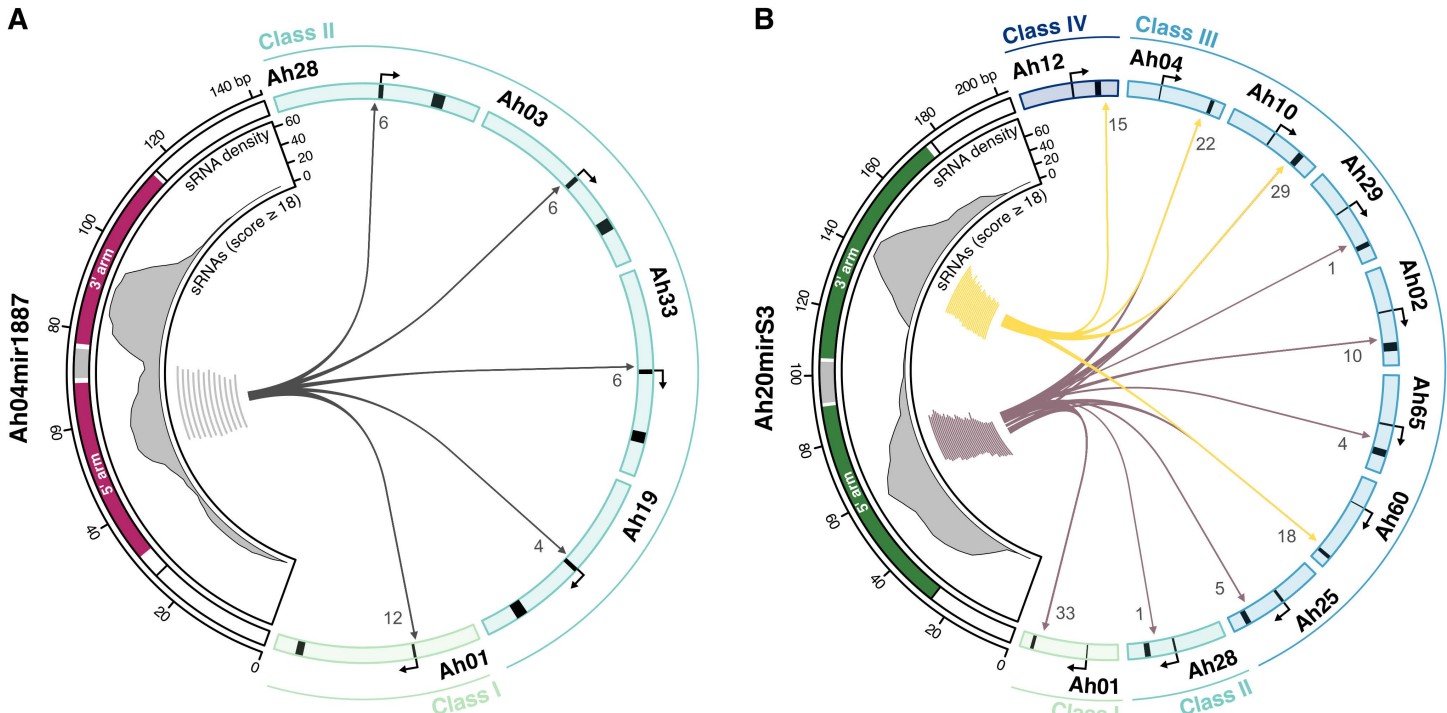

**Fig 5. The targeting repertoire of *S*-locus sRNA precursors is enhanced by their heterogeneous processing pattern.** Circos plot representing the positional relationship between sRNAs produced by Ah04mir1887 **(A)**, Ah20mirS3 **(B)** and the corresponding targeted *S*-alleles. The genomic regions harboring Ah04mir1887 and Ah20mirS3 are represented on the left side of the plots. The predicted 5' and 3' hairpin arms are represented in magenta (Ah04mir1887) and green (Ah20mirS3), and the sRNA density (number of sRNAs) is plotted along the sRNA precursor region. Individual sRNAs with a score of ≥ 18 are represented by the grey, yellow or dark purple segments. Arrowed segments represent the sRNA source - target relationship. The numbers at the end of these arrowed segments indicate the total number of sRNAs targeting that respective allele. The *SCR* genomic region of targeted *S*-alleles is represented on the right side and includes 1500 bp upstream of the TSS and 500 bp downstream of the stop codon. *SCR* exons 1 and 2 are colored in black.

in the nucleus, which is accompanied by accumulation of siRNAs of the cleaved pre-mRNA [41]. Additionally, all main miRNA processing components, including AGO1, are present in the nucleus [42–44]. Thus, given the independent slicer activity of AGO1 [45], it could be hypothesized that this ARGONAUTE uses sRNAs to target the *SCR01* pre-mRNA in the nucleus, promoting its cleavage co-transcriptionally. Alternatively, the interaction between AGO1 and the *SCR* pre-mRNA could promote disassembly of the Pol II complex leading to the arrest of transcription, as has been suggested to occur in some miRNA genes upon salt stress [46]. Even though we could not detect any sRNAs corresponding to the cleavage of the *SCR* (pre-)mRNA, nor secondary siRNAs indicative of transitivity [47] at this locus in any of our sRNA sequencing experiments (S11 Fig), future experiments should investigate the possibility that the sRNA precursors at the *S*-locus could act through nuclear PTGS or through transcriptional arrest.

Although we caution against a literal interpretation of the BSAS DNA methylation data due to potential technical limitations that would require cell-type-specific sorting to overcome, our molecular and phenotypic results together suggest that S-locus sRNA precursors function independently of DNA methylation and the RdDM pathway, in contrast to previous observations in *Brassica*: in this latter system, the sRNA precursors *Smi* and *Smi2* are associated with deposition of DNA methylation in and around the *SCR* target site and a decrease in *SCR* expression [12,20,21]. This could be a result from a direct incorporation of *Smi* and *Smi2* sRNAs into AGO4-clade proteins which would elicit *de novo* DNA methylation through the RdDM pathway, as has been observed for selected miRNAs and inverted repeats [23–27]. Alternatively, as suggested by Finnegan et al. [48], it could depend on *Smi*-induced cleavage of an antisense transcript at the *SCR* locus, generating sRNAs that could subsequently be co-opted into the RdDM pathway, although this type of transcript is hypothetical and has never been reported in Brassica or found in our own sequencing of Arabidopsis samples (S1 Fig). Even though empirical investigations are still necessary to test these hypotheses in Brassica, the cumulative evidence from these earlier studies and our own suggests that different sRNA precursors within the Brassicaceae self-incompatibility system can recruit distinct effector pathways, that nevertheless converge into the same molecular phenotype: reduction of *SCR* transcript abundance. This would imply that these regulatory elements have an outstanding molecular flexibility, despite functioning within the specific evolutionary constraints of the self-incompatibility system. Thus, it cannot be excluded that different *A. halleri* sRNA precursors, as well as precursors from other self-incompatible species such as *A. lyrata* and *Capsella grandiflora* could use different gene regulatory pathways. Future investigations of different *S*-locus sRNA precursor families in these species, using heterologous expression systems as done in this study, or in the native context whenever genetic transformation is feasible, will hopefully help answer this outstanding question.

## The complex *S*-allele dominance network relies on the unique features of S-locus sRNA precursors

Durand et al. [19] previously showed that the *S*-locus sRNA precursors of *A. halleri* have high generalism, and suggested that dominant *S*-alleles in particular, rely on this feature to target multiple recessive alleles, rather than on carrying an increased number of sRNA precursors. Our study uncovers the molecular basis for this generalism, and shows that the apparent haphazard processing of *S*-locus sRNA precursors is required to generate numerous functional sRNAs that target multiple alleles cooperatively: remarkably, all *SCR* alleles analyzed in this study are targeted by multiple sRNAs of a single sRNA precursor, and/or are targeted by distinct sRNA precursors. This contrasts with the less complex dominance hierarchy of Brassica, where fewer allelic interactions are present, and a single functional sRNA molecule per precursor has been suggested to mediate all dominance relationships [20,21]. It is important to note that while we investigated a total of 15 *S*-alleles in this study, at least 43 are known to exist in *A. halleri* [7], and > 80 in the closely related *Capsella grandiflora* [14]. Therefore, in this study we most likely underestimated the fraction of functional sRNAs in each precursor, since those that are non-functional in the allelic interactions explored here could potentially have a role in interactions that have not yet been characterized.

Overall, the evidence suggests that collaborative targeting, either by multiple sRNAs of a single precursor, or by sRNAs from distinct sRNA precursors, is an essential feature of this system: firstly, it is essential for granting an *S*-allele its complete targeting spectrum against the many *S*-alleles present in the population; and secondly, the use of multiple sRNAs with distinct molecular features could trigger diverse effector pathways acting on the same target allele, which could perhaps underlie the robust reduction of recessive *SCR* transcripts observed in all allelic interactions studied thus far (this study and [10]).

Intriguingly, this resembles the collaborative non-self recognition model observed in Solanaceae and other plant families, where each *S*-locus allele contains multiple *SLF* genes [49]. In these systems individual SLF proteins neutralize multiple non-self S-RNases that would otherwise inhibit pollen germination. This generalism ensures broad recognition and maintains complex inter-allelic interactions [49]. Although the underlying molecular mechanisms differ, a conceptual parallel exists in *A. halleri*: here, generalism arises from the sRNA precursor's ability to produce numerous sRNAs with

distinct targeting spectra, rather than from multiple protein-protein interactions as in the SLF–S-RNase system [50]. This parallel illustrates how different molecular mechanisms can converge on the same functional outcome - ensuring broad recognition and maintaining complex dominance relationships within self-incompatibility systems.

### *S*-locus sRNA precursors provide a window into the miRNA evolution continuum

Although sharing some structural similarities with miRNAs, the features of *S*-locus sRNA precursors more closely resemble those of proto-miRNAs [51]. Proto-miRNA loci are inverted repeats present in several plant genomes that have acquired transcriptional competence, forming long hairpins which can be diced by multiple Dicer-like proteins (DCLs), leading to a complex and heterogeneous population of sRNAs [51–55]. There is little evidence that such a sRNA population could cause substantial effects in gene expression, at least through the canonical PTGS pathway; therefore, these structures have been mostly regarded as substrates for selection to act on, rather than having immediate biological significance [51–55]. We hypothesize that natural selection promotes diversification of the targeting spectrum of these *S*-locus sRNA precursors towards a broader set of recessive *S*-alleles, rather than towards highly specific interactions as seen in canonical miRNAs [56]. This selective pressure likely contributed to the persistence of proto-miRNA features over extended evolutionary timescales, preventing their maturation into canonical miRNAs, which would likely compromise their ability to target multiple *S*-alleles, and thus the structure of the dominance network. Our previous work has shown that *S*-locus sRNA precursors vary widely in evolutionary age: some, such as mirS4 and mir867, have emerged relatively recently, while others trace back to the diversification of the Brassicaceae family approximately 30 Mya [19]. It is therefore clear that these older sRNA precursors, including mirS3 and mir1887 studied here, have retained proto-miRNA features over long evolutionary periods. It would be interesting to test if apparently stabilized proto-miRNAs are also used in other biological systems where the expression of multiple genes/alleles needs to be coordinated by a single regulator and under similar evolutionary constraints as those acting on the *S*-locus. The mimicry of wing patterns in *Heliconius* butterflies would be an interesting model system to test this, since balancing selection has favored the appearance of numerous alleles of the supergene controlling wing pattern, and dominance relationships between them are prevalent [57,58].

## Materials and methods

### Plant material & growing conditions

Wild type (WT) and transgenic *Arabidopsis thaliana* plants used in this study were grown in a greenhouse, using standard growth conditions (16 h light/8 h dark; 110 µmol/s/m$^2$; 21°C; 70% relative humidity). Seeds were germinated directly on soil, or in MS-medium (0.43% MS-salts, 0.8% Bacto agar, 0.19% MES hydrate, and 1% sucrose) when antibiotic selection was necessary for transgene selection.

Because the self-incompatibility reaction cannot be faithfully reconstructed in the *A. thaliana* Col-0 ecotype [28–31], all wild type and mutant plants used in this study were of the C24 ecotype. *polIV* and *polV* correspond to the previously published and validated *rdm5 (nrpd1a)* and *rdm6 (nrpd1b)* mutant alleles [33]. These two mutants were generated in the *ros1* mutant background, which was removed in our study by backcrossing to WT C24 plants. The *ago4 ago6* double mutant was generated in the C24 background using CRISPR/Cas9 (see section *Generation and validation of CRISPR/Cas9 mutants* for more details).

To obtain the *S*-allele sequences of Ah65, Ah60, Ah03, Ah33 and Ah19, five *Arabidopsis halleri* individuals containing a combination of these alleles were collected from diverse natural European populations [59], and grown under standard greenhouse conditions in order to obtain plant material for DNA extraction and sequencing (see the section *DNA extraction, Nanopore sequencing and assembly of S-alleles* for more details). Following Goubet et al. [15], we further obtained the *S*-allele sequences of Ah01, Ah02 and Ah25 by constructing and screening BAC libraries from fresh leaf samples, and fully sequencing the positive clones using PACBIO.

## Cloning and generation of transgenic plants

Transgenic lines carrying *SRK01*, *SCR01*, and the sRNA precursor Ah20mirS3 were previously published [19]. The precursors Ah20mir1887 and Ah20mirS3 are in very close physical proximity within the Ah20 allele, and the previously published Ah20mirS3 line contains both sRNA precursors [19]. While performing sRNA sequencing of AGO-IP experiments (see section *ARGONAUTE immunoprecipitation and sRNA sequencing*), we noticed that some Ah20mir1887 sRNAs show homology to the 5' region of *SCR01*. The base-pairing score of these predicted interactions varies between 12.5 and 18.5, depending on the sRNA (see the section *sRNA database construction and target site inference for Ah04mir1887 and Ah20mirS3* for more details on these scores). To confidently isolate the effect of Ah20mirS3 in the dominance interaction of Ah20 over Ah01 (Interaction B), we introduced five point mutations in the *SCR01*^*mir1887* sequence that disrupt the homology with the Ah20mir1887 sRNAs, rendering the Ah20mir1887 target site non-functional. We chose this option in order to maintain the native locus context around Ah20mirS3 which might contain regulatory sequences necessary for correct expression and processing of this sRNA precursor. The *SCR01*^*mir1887* transgene was generated by performing site-directed mutagenesis on the already published *SCR01* clone [19], using the primers detailed in S1 Table. The resulting amplicon was then recombined into the entry vector pDONR-Zeo, and subsequently recombined into the pB7m34GW destination vector using the 3-fragment Gateway Cloning System (Invitrogen), including 5' and 3' mock sequences, as detailed previously [19].

   Ah04mir1887 was cloned by amplifying a region spanning this sRNAs precursor and containing 2 kb of upstream and downstream sequences from a BAC clone carrying allele Ah04, using the primers detailed in S1 Table. The resulting amplicon was then recombined into the entry vector pDONR-Zeo, and subsequently recombined into the pH7m34GW destination vector using the 3-fragment Gateway Cloning System (Invitrogen), including 5' and 3' mock sequences, as detailed previously [19].

   *Arabidopsis thaliana* C24 plants were transformed using the floral dip method [60], and all transgenic lines were selected as described in Durand et al. [19]. Briefly, for *SRK01,* a single-insertion reference line was selected from multiple independent transgenic lines displaying a stable and reproducible SI phenotype [19]. This previously characterized line was used throughout the present study. A similar selection strategy was applied to *SCR* constructs (*SCR01*, previously isolated and characterized in Durand *et al.* [19], and *SCR01*^*mir1887*, produced in this study): multiple independent *SCR* lines were generated and tested, all of which exhibited comparable SI phenotypes when crossed with the previously established *SRK01* reference line. For each construct, one representative single-insertion line was selected and used as the reference base line to generate all pollen donor genotypes used in this study. For lines expressing sRNA precursors (Ah20mirS3, previously isolated and characterized in Durand *et al*. [19], and Ah04mir1887, produced in this study), multiple independent lines were generated, and precursor expression was confirmed by qPCR. These lines were introgressed into the corresponding *SCR* reference background, and one representative line showing abolition of the SI phenotype when crossed to the *SRK01* reference line was selected as the reference. This line was subsequently introgressed into the RdDM mutants of interest to generate hemizygous lines carrying either a single copy of *SCR* alone, or a single copy of *SCR* together with the sRNA precursor, in either wild-type or mutant genetic backgrounds. These hemizygous lines were used as pollen donors in the phenotypic SI assays and for qPCR analyses of *SCR01* expression.

## Generation and validation of CRISPR/Cas9 mutants

Given the aforementioned limitation that the self-incompatibility reaction of *A. halleri* can only be reconstituted in the *A. thaliana* C24 ecotype [28–31], we created the *ago4 ago6* RdDM mutants in this specific ecotype using CRISPR/Cas9. The expression cassette of pCBC-DT1T2 [61] was amplified using two overlapping forward and reverse primers that contained two gene-specific sgRNA sequences (Fw1: 5'-ATATAT<u>GGTCTC</u>GATTGNNNNNNNNNNNNNNNNNNNNGTT-3'; Fw2: 5'-TGNNNNNNNNNNNNNNNNNNNNGTTTTAGAGCTAGAAATAGC-3'; Rv1: 5'-ATTATT<u>GGTCTC</u>GAAACNNNN NNNNNNNNNNNNNNNNCAA-3'; Rv2: 5'-AACNNNNNNNNNNNNNNNNNNNNCAATCTCTTAGTCGACTCTAC-3', where

N corresponds to the gene-specific gRNAs (S1 Table), and underlined regions correspond to BsaI restriction sites). The amplicons obtained from this PCR reaction were gel-isolated and inserted into pHEE401E [62] using BsaI and T4 ligase (Thermo Fisher Scientific). The pHEE401E vectors containing gRNAs against *AGO4* and *AGO6* were transformed into the *Agrobacterium tumefaciens* strain GV3101. The floral-dip method [60] was then used to transform WT C24 *A. thaliana* plants.

To screen for T1 mutant plants, the regions targeted by the gRNAs were Sanger sequenced to identify *ago4 ago6* mutants with premature stop codons in both genes (S3 Fig). In the T2 generation, double homozygous mutants that did not carry the pHEE401E were selected by genotyping (see S1 Table for primers).

To confirm that *ago4 ago6* mutations impair DNA methylation, a Chop-PCR assay was performed [63]. Briefly, this assay uses genomic DNA treated with the methylation-sensitive enzyme HaeIII as a template for PCR. Primers targeting regions known to have AGO4/AGO6-dependent DNA methylation, as well as control unmethylated regions are amplified (see S1 Table for primers). Using this method, we could determine that the *ago4 ago6* mutant generated in this study shows reduced DNA methylation levels, as has been previously published for other mutant alleles of these genes (S3C Fig) [64].

### Pollen germination assays

All lines used for the reconstruction of the self-incompatibility reaction in *A. thaliana* were used in hemizygous state to mimic the abundance of sRNA precursors, and *SCR/SRK* in *A. halleri* individuals heterozygous at the *S*-locus. Genotyping of each transgenic line/mutant was done by classical PCR, or using the KASPar assay [65] (S1 Table). Paternal plants hemizygous for *SCR01* and sRNA precursors were obtained by crossing *SCR01* to Ah04mir1887 homozygous lines or by crossing *SCR01*<sup>*mir1887</sup> to Ah20mirS3 homozygous lines.

To assess the self-incompatibility reaction at the phenotypic level, pollen germination assays were performed on manually crossed plants. Plants hemizygous for *SRK01* were emasculated one day before anthesis, and manually pollinated 24 h later with pollen derived from plants hemizygous for *SCR01, SCR01*<sup>*mir1887</sup>, *SCR01* Ah04mir1887 or *SCR01*<sup>*mir1887</sup> Ah20mirS3. Pistils were pollinated using an excess of pollen from dehiscent anthers, in order to saturate the pistil and to ensure that low pollen germination counts would not be due to absence of pollen grains. Pollinated pistils were transferred to fixing solution 6 h after pollination and stained with aniline blue, as described before [19]. These pistils were then mounted on microscope slides and imaged using a Zeiss AX10 fluorescence microscope, where we counted the number of germinated pollen grains present in each pistil (S7 Fig). When a pistil carried more than 30 germinated pollen grains the counting was capped at 30 since this number of germinated pollen grains is indicative of a robust compatible reaction [19].

To evaluate the effect of RdDM mutants on the activity of *S*-locus sRNA precursors we crossed the RdDM mutants *ago4 ago6*, *polIV* and *polV* with the previously described paternal plants carrying *SCR01* and sRNA precursors (*SCR01, SCR01*<sup>*mir1887</sup>, *SCR01* Ah04mir1887 and *SCR01*<sup>*mir1887</sup> Ah20mirS3). We then obtained plants hemizygous for the *SCR01* and sRNA precursor transgenes and homozygous for the mutations of interest and used these as paternal plants in crosses with maternal hemizygous *SRK01* plants, as described above.

Pollen germination differences were assessed using a two-sided Wilcoxon rank-sum test comparing the presence versus absence of sRNA precursors in each mutant background. As a control, all maternal and paternal lines used in this study were crossed to WT C24. Germination and seed set for all these genotypes were comparable to that of WT x WT crosses, confirming that the transgenes and mutations had no detectable effect on plant fertility.

### Phenotypic determination of the *S*-loci dominance hierarchy

The *S*-locus dominance hierarchy in Fig 3 is based on a compilation of controlled reciprocal crosses between different *A. halleri* individuals [10,19,37,66]. In cases where interactions between alleles of two different dominance classes had not been tested phenotypically, dominance was inferred here based on the phylogenetic class of each allele, since interclass

dominance interactions are predictable [7,16]. In the case of intraclass relationships, such as for alleles Ah65, Ah60 and Ah33 (all belonging to class III), phenotypic dominance has not been determined yet, and thus the ordering of these alleles along the hierarchy remains arbitrary at this stage.

## RT-qPCR

To measure the abundance of *SCR01* transcripts, we isolated immature buds (stage 8–11 according to Smyth et al. [67]) of plants hemizygous for *SCR01, SCR01*$^{*mir1887}$*, SCR01* Ah04mir1887 or *SCR01*$^{*mir1887}$ Ah20mirS3 in wild-type background, or in the indicated mutant backgrounds (*ago4/- ago6/-, polIV/-* and *polV/-*). The individual plants used for RNA extraction were the same as those used as pollen donors in the SI phenotypic assays shown in Fig 1. Accordingly, pollen donors used for the germinated pollen phenotype panel correspond directly to the plants used for the *SCR01* transcript abundance analysis, and similarly colored dots in both panels represent the same individual plants. Each individual plant is considered a biological replicate.

RNA was extracted using the NucleoSpin RNA Plus kit (Macherey-Nagel) and cDNA was synthesized with the RevertAid First Strand cDNA Synthesis Kit (Thermo Scientific). qPCR was performed using the iTaq Universal SYBR Green Supermix (BioRad) in a Lightcycler 480 instrument (Roche), with the primers found in S1 Table. *ACT8* was used as a reference gene. Transcript abundance was quantified using the Pfaffl method [68]. Expression values were normalized to the *ACT8* reference gene and averaged across 3 technical replicates to obtain one value per biological replicate. Normalized expression values were log2-transformed prior to statistical analysis (S2 Table). Differences between control and samples within each genotype were assessed using an unpaired two-tailed Welch's t-test to account for unequal variances ($n = 4$ biological replicates per group) (S3 Table).

## ARGONAUTE immunoprecipitation and sRNA sequencing

AGO1 and AGO4 immunoprecipitation (IP) assays were performed on immature buds (stage 11 and younger, according to Smyth et al. [67]) of *A. thaliana* plants carrying homozygous Ah04mir1887 or Ah20mirS3 transgenes, using a previously published protocol [69], with the following modifications: IPs were performed on 1.4 g of finely ground powder of frozen immature buds. The anti-AGO1 (AS09 527, Agrisera) and anti-AGO4 (AS09 617, Agrisera) antibodies were added at a 1/500 dilution for the incubation step and their immobilization was performed using Dynabeads Protein G (Invitrogen). The IP products were washed six times with PBS prior to sRNA extraction using the Trizol reagent (Ambion). In parallel, an aliquot of each input extract was used for total sRNA isolation using the Trizol LS reagent (Invitrogen), according to the supplier instructions. For each genotype three sRNA fractions were obtained: AGO1-IP, AGO4-IP and input.

These three sRNA fractions were subjected to an acrylamide gel-based size selection for sRNAs. These sRNAs were then used for TruSeq Small RNA (Illumina) library preparation and sequenced on a NextSeq 500 platform (Illumina), using 75 bp single-reads. Reads were trimmed, quality filtered, and size-selected using TrimGalore [70]. We followed a two-step mapping procedure with ShortStack (allowing for one mismatch) [71], where we mapped sRNAs from each sample to the *A. thaliana* TAIR10 genome (masking the *S*-locus region), and to the sequence of the corresponding *A. halleri S*-allele (Ah04, for samples carrying Ah04mir1887 and Ah20 for samples carrying Ah20mirS3). Only reads that mapped exclusively to the *S*-locus were kept for further analysis. sRNA read counts were expressed as rpm to allow comparison of abundance levels across samples.

AGO loading was inferred based on the ratio of reads found in the AGO1-IP fraction and the AGO4-IP fraction: a sRNA was considered to be loaded in AGO1 or AGO4 when more than 50% of its total reads were derived from either the AGO1-IP fraction or the AGO4-IP fraction, respectively. A given sRNA was classified as having an unknown loading when no reads were found in both AGO-IP fractions, or when the amount of reads found in the input fraction was superior to that found in any of the two AGO-IP fractions.

## sRNA database construction and target site inference for Ah04mir1887 and Ah20mirS3

To identify sRNA target sites within *SCR* genomic regions, we first compiled a comprehensive database incorporating all sRNAs sequenced in the AGO-IP experiments described above, along with previously sequenced sRNAs [19]. This database represents the complete set of sRNAs sequenced to date in our lab, originating both from *A. thaliana* and *A. halleri* plants carrying Ah04mir1887 or Ah20mirS3. To validate the quality of the sRNA sequencing experiments used to generate this database, we assessed characteristic features of sRNAs mapping to well-defined regulatory regions, including canonical miRNAs, TAS loci, TEs, and *de novo* RdDM loci (S12 Fig). These analyses showed the expected size distributions, 5′ nucleotide biases, and AGO associations for each class, confirming the reliability of these sRNA datasets. In parallel, a database containing *SCR* genomic regions of different *A. halleri S*-loci was compiled. This included alleles from all four dominance classes, and comprised previously published sequencing data (see *Data Availability* section for a complete list of data sources). In addition to this, long read Nanopore sequences newly obtained from individuals carrying alleles Ah65, Ah60, Ah03, Ah33 and Ah19 were also added to this database (see section *DNA extraction, Nanopore sequencing and assembly of S-alleles* for more details).

Using the databases of sRNA sequences and *SCR* alleles, we applied a previously published homology detection algorithm [19] to find target sites. Briefly, sRNA targets were predicted using the Smith-Waterman algorithm with matches scored +1, mismatches −1, gaps −2, and G:U wobbles −0.5.

Functional sRNAs are defined as those with a homology score of ≥ 18. In prior work, *SCR* transcript repression was consistently observed for interactions scoring ≥ 18, with an abrupt transition between high and low *SCR* expression around this value, supporting the use of a hard threshold rather than a gradual quantitative model [10,19]. The identified sRNA - *SCR* interactions were then represented in a heatmap (Fig 4) using ComplexHeatmap [72], and in a circos plot (Fig 5) using circlize [73].

## DNA extraction, Nanopore sequencing and assembly of *S*-alleles

To obtain the *S*-locus sequences of alleles Ah65, Ah60, Ah03, Ah33 and Ah19, 2 g of fresh leaves were collected from five different individuals carrying these alleles and flash-frozen. High molecular weight genomic DNA was extracted as described before [74]. Long read sequences were obtained by Oxford Nanopore Technologies (ONT) flowcells v9. To polish the contigs obtained by long read sequencing Illumina data were also generated (with no step of PCR amplification to minimize sequencing bias).

For Nanopore library preparation, the smallest genomic DNA fragments were first eliminated using the Short Read Eliminator Kit (Pacific Biosciences). Libraries were then prepared according to the protocol 1D Native barcoding genomic DNA (with EXP-NBD104 and SQK-LSK109) provided by Oxford Nanopore Technologies. Depending on how many samples were pooled, 250 ng (pool of 9 samples) to 1 µg (pool of 4 samples) of genomic DNA fragments were repaired and end-prepped with the NEBNext FFPE DNA Repair Mix and the NEBNext Ultra II End Repair/dA-Tailing Module (New England Biolabs). Barcodes provided by ONT were then ligated using the Blunt/TA Ligase Master Mix (NEB). Barcoded fragments were purified with AMPure XP beads (Beckmann Coulter), then pooled and ONT adapters were added using the NEBNext Quick Ligation Module (NEB). After purification with AMPure XP beads (Beckmann Coulter), each library was mixed with the sequencing buffer (ONT) and the loading beads (ONT) and loaded on a PromethION R9.4.1 flow cell. In order to maintain the translocation speed, flow cells were refueled with 250 µl Flush Buffer when necessary. Reads were basecalled using Guppy version 3.2.10, 4.0.11, 5.0.12, 5.0.13, 5.0.16 or 5.1.12. The Nanopore long reads were not cleaned and raw reads were used for genome assembly.

For Illumina PCR-free library preparation, 1.5 µg of genomic DNA was sonicated to a 100–1500-bp size range using a Covaris E220 sonicator (Covaris). The fragments (1 µg) were end-prepped, and Illumina adapters (NEXTFLEX Unique Dual Index Barcodes, Perkin Elmer) were added using the Kapa Hyper Prep Kit (Roche). The ligation products were purified twice with 1X AMPure XP beads (Beckman Coulter). The libraries were then quantified by qPCR using the KAPA

Library Quantification Kit for Illumina Libraries (Roche), and their profiles were assessed on an Agilent Bioanalyzer (Agilent Technologies). The libraries were sequenced on an Illumina NovaSeq 6000 instrument (Illumina) using 150 base-length read chemistry in a paired-end mode.

Nanopore sequencing data were assembled using Necat [75] with a genome size of 240 Mbp and the remaining parameters left as default. Contigs produced by Necat were polished one time using Racon [76] with Nanopore reads, then one time with Medaka (https://github.com/nanoporetech/medaka, model r941_prom_hac_g507) and Nanopore reads, and two times with Hapo-G v1.3.4 [77] and Illumina short reads. However, the cumulative sizes of our long-read assemblies were larger than the estimated 240 Mb, suggesting that the assembly size was currently inflated by the presence of allelic duplications. Thus, HaploMerger2 [78] was run (Batch A twice to remove major misjoin and one Batch B) to generate a haploid version of each assembly.

To detect the *S*-locus region in the generated assemblies, blast searches were performed using the genes flanking the *S*-locus (*ARK3* and *U-box* [15]) as queries. This allowed us to precisely narrow down the genomic regions corresponding to the full Ah65, Ah60, Ah03, Ah33 and Ah19 *S*-alleles. We annotated *SRK* using blast searches, and following Goubet et al. [15], we used Fgenesh+ [79] to predict the genomic sequence of *SCR* based on a database of SCR protein sequences from already known *S*-alleles.

### Determination of the *SCR01* transcription start site (TSS)

To identify the TSS of *SCR01* in *A. halleri*, total RNA was extracted from unopened buds of three individual plants from accession Zapa12 (Central Europe), Bara01 and Firi03 (Romania) [59], genotyped as homozygous for the Ah01 allele. A library was directly prepared using the NEBNext Ultra II Directional RNA Library Prep for Illumina Kit. Libraries were sequenced with an Illumina NovaSeq 6000 instrument (Illumina) using a paired-end 151 bp read chemistry. Short Illumina reads were bioinformatically processed to remove adapters and primer sequences, only reads ≥ 30 nucleotides were retained. These filtering steps were done using in-house-designed software based on the FastX package [80]. Finally, read pairs that mapped to ribosomal sequences were filtered using SortMeRNA as previously described [81]. Processed reads were aligned with STAR [82] using –alignIntronMax 3000 and –alignMatesGapMac 10000 parameters to avoid very long splicing assignment errors due to a mammalian-based design of the STAR algorithm. The vertical coverage at each base pair for the three replicates was computed using samtools depth [83] and then plotted in Rafter feature scaling and smoothing the curves using mean coverage values in 20 bp sliding windows.

### BSAS sequencing

To measure DNA methylation in *SCR01* target sites we used the BiSulfite Amplicon Sequencing Technique - BSAS [26,35], which relies on PCR enrichment of regions of interest using bisulfite converted DNA as a template. To assess if the presence of *S*-locus sRNA precursors is associated with DNA methylation changes in the respective *SCR01* target regions, we collected immature buds (stage 8–11 according to Smyth et al. [67]) of *SCR01, SCR01*[*mir1887*], *SCR01* Ah04mir1887 and *SCR01*[*mir1887*] Ah20mirS3 plants. Eight to ten biological replicates were extracted and bisulfite-converted for each genotype. In parallel with bud collection, SRK01 pistils were pollinated with the anthers from the same plants to confirm the phenotypic self-incompatibility response of the collected lines. Crosses and pollen germination measurements were performed as described previously (see section *Pollen germination assays*). The DNA methylation and phenotypic measurements are paired, meaning that the same individual plant was assayed for both aspects; these are represented by dots of the same color in S5C and S5D Fig.

Genomic DNA was extracted using the NucleoSpin Plant II kit (Macherey-Nagel), and converted with the EZ DNA Methylation Kit (Zymo Research), including the modifications A and B, as described by the supplier, to optimize the conversion efficiency. Converted DNA was then amplified using primers flanking the Ah04mir1887 or Ah20mirS3 target sites in the intron and 5' region regions of *SCR01*, respectively (S1 Table). For each sRNA precursor, seven amplicons were

generated, which amplify the top and bottom strands of the ~800 bp that flank the target site. As a negative control, we also amplified *AT2G20610*, which has been previously described to be an unmethylated negative control in *A. thaliana* Col-0 plants [26]. Amplification of bisulfite converted DNA was performed using the TaKaRa EpiTaq HS (Takara), and the resulting PCR reactions were cleaned-up using the NucleoSpin Gel and PCR Clean-up XS kit (Macherey-Nagel). All amplicons obtained from a single biological replicate were pooled in equimolar amounts and a total of 100 ng were used for library preparation following the modified Illumina Nextera Flex library preparation protocol described before [26,84]. Libraries were sequenced in a MiSeq (Illumina) platform using 250 bp paired-end reads. Reads were trimmed, quality-filtered, and mapped to the Ah01 allele using methlypy [85], following the paired-end pipeline and specifying *AT2G20610* as the unmethylated control. Conversion efficiencies for each sample were estimated based on the methylation levels detected in the *AT2G20610* locus, and were between 97% and 83% (S4 Table). The number of methylated reads covering each *SCR01* cytosine position, as givenby methylpy, can be found in S5 Table. Average DNA methylation levels within the top and bottom strands of the ~800 bp regions flanking Ah04mir1887 and Ah20mirS3 target sites were summarized in R [86] and plotted with ggplot2 [87].

To identify DNA methylation heterogeneity at the single-molecule level, the methylation rate over each single read aligned to the *SCR01* sequence in three different cytosine contexts (CG, CHG and CHH) was computed using the Bismark methylation call output [88]. For each sample, reads were classified in three distinct categories: highly methylated reads with 90% or more methylated cytosines, weakly methylated reads with a methylation rate inferior to 10%, and everything else in between. Statistical differences in the numbers of highly and weakly methylated reads between *SCR01* and *SCR01* Ah04mir1887 or *SCR01*[*mir1887*] and *SCR01*[*mir1887*] Ah20mirS3 genotypes were assessed using a two-tailed Mann-Whitney test with continuity correction.

## Supporting information

**S1 Table. Primer sequences used in this study.**
(XLSX)

**S2 Table. RT-qPCR quantification of *SCR01* expression in Interaction A and Interaction B.**
(XLSX)

**S3 Table. *SCR01* expression differences between genotypes (log2FC/ p-value/ significance level).** Differences were assessed using an unpaired two-tailed Welch's t-test on the log2-transformed values in S2 Table.
(XLSX)

**S4 Table. Estimated bisulfite conversion efficiencies of BSAS samples.**
(XLSX)

**S5 Table. Methylpy output containing number of methylated reads in *SCR01* and AT2G20610 regions.**
(XLSX)

**S6 Table. Raw sRNA read counts at different control loci, for plants expressing Ah04mir1887 or Ah20mirS3.**
(XLSX)

**S1 Fig. *SCR01* TSS in relation to Ah04mir1887 target site.** Normalized RNA-seq coverage along the *SCR01* gene is shown for three biological replicates of *Arabidopsis halleri* plants homozygous for Ah01. Exons and introns are represented by black and white regions, respectively. The Ah04mir1887 target site is highlighted in pink, and the three replicates are depicted as curves in different shades of blue. Coverage values (number of reads aligned at each position) were normalized to the maximum value for each replicate to allow comparison of transcription profiles. Curves were smoothed using a 20 bp sliding window with a 1 bp step. The bottom panel provides a magnified view of the target site, showing

individual mapped reads from the replicate with the highest coverage relative to the start of Exon 1. Tick marks on the x-axis indicate the distance in base pairs (bp) from the translation start site. This detailed view shows that the majority of *SCR01* transcripts initiate immediately downstream of the Ah04mir1887 target site, approximately -30 to -10 bp from the translation start site, suggesting that the targeted sequence is not included in the most abundant *SCR01* transcript isoform.
(TIFF)

**S2 Fig. *SCR01*<sup>\*mir1887</sup> transgenic line.** (A) Alignment between the native mir1887 target site on *SCR01* and Ah20mir1887 sRNAs with a targeting score ≥ 18. (B) Alignment between the same sRNAs and the mutated mir1887 target site of the transgenic line *SCR01*<sup>\*mir1887</sup>. Note that the mutations decrease the homology score below 18, suggesting that these sRNAs are not able to target the mutated site. Mutated sites are in red.
(TIFF)

**S3 Fig. Validation of the RdDM CRISPR/Cas9 mutants generated in this study. (A-B)** Schematic representation of CRISPR/Cas9 induced mutations in the *AGO4* and *AGO6* genes. Sequence changes are shown in red, premature stop codons are represented by the asterisk. **(C)** Chop-PCR assay on WT C24 plants and *ago4 ago6* double mutant plants. *IGN23* and *AtSN1* correspond to RdDM-dependent methylated regions, while *RDRP* corresponds to a non-methylated region.
(TIFF)

**S4 Fig. A compatible pollen germination phenotype requires a low *SCR01* transcript abundance.** Non-linear regression analysis modeling the relationship between *SCR01* transcript abundance and pollen germination. For both Ah04mir1887 **(A)** and Ah20mirS3 **(B)** sRNA precursors, a compatible pollen germination phenotype (~20–30 germinated pollen grains) requires approximately a 10-fold decrease in SCR01 transcript abundance compared to wild-type levels. Modelling was performed using the SSasymp self-starting asymptotic regression model in R. This analysis was based on the phenotypic and expression data presented in Fig 1C and 1D. *SCR01* expression was measured once in each pollen donor, while pollen germination was recorded on multiple pistils pollinated with the same pollen donor. To account for this non-independence, germination values were averaged per plant prior to model fitting. The black line represents the predicted model, and the gray area corresponds to the 95% confidence interval.
(TIFF)

**S5 Fig. BiSulfite Amplicon Sequencing (BSAS) of *SCR01* regions targeted by Ah04mir1887 and Ah20mirS3. (A)** Schematic representation of the BSAS assay performed to measure DNA methylation at the Ah04mir1887 target site (*SCR01* 5' region - Region A), and at the Ah20mirS3 target site (*SCR01* intron - Region B). Region A and Region B correspond to fragments of 794 and 871 bp, respectively, centered in each sRNA precursor's target site. **(B)** Phenotypic validation of lines used for the BSAS assay. Colored dots correspond to values obtained for each biological replicate. Number of biological replicates is indicated at the top of the panel. For each biological replicate pollen germination on a total of 5 *SRK01* pistils was counted. Statistical differences were assessed using a two-tailed Mann-Whitney test with continuity correction **(C)** Average DNA methylation levels on Region A (Interaction A) and Region B (Interaction B) of *SCR01*. DNA methylation levels correspond to the average of methylation in all cytosines in the top and bottom DNA strands of sampled regions. Measurements are paired, with each plant used for both pollination (panel B) and the BSAS experiment (panel C) represented by the same color dot, allowing a direct comparison of phenotypic response and DNA methylation levels for each individual. The number of biological replicates is indicated at the top of the panel. Statistical differences were assessed using a two-tailed Mann-Whitney test with continuity correction (ns - non significant, p-value > 0.05; ** - p-value ≤ 0.01).
(TIFF)

**S6 Fig. DNA methylation levels at Ah04mir1887 and Ah20mirS3 target sites on *SCR01*.** BSAS reads were grouped according to the fraction of methylated cytosines within each read. High: reads where nearly all cytosine positions are methylated (> 90% cytosines); Low: reads where very few cytosine positions are methylated (< 10% cytosines); Medium: reads with more than 10% and less that 90% cytosines methylated. Error bars represent the standard deviation between replicates. No statistically significant differences were detected in the number of highly, medium and lowly methylated reads between the *SCR01* and *SCR01* Ah04mir1887 or *SCR01*[*mir1887] and *SCR01*[*mir1887] Ah20mirS3 genotypes (p-value > 0.05, two-tailed Mann-Whitney test with continuity correction).
(TIFF)

**S7 Fig. Aniline blue staining of pistils showing incompatible and compatible pollen germination reactions.** Pistils were pollinated with incompatible (left) or compatible (right) pollen and stained with aniline blue. The right panel shows strong fluorescence corresponding to germinating and growing pollen tubes within the style, characteristic of a compatible pollen reaction. In contrast, incompatible pollen (left) fails to germinate or produce elongating pollen tubes. Scale bar: 100μm.
(TIFF)

**S8 Fig. sRNAs produced from Ah04mir1887 and Ah20mirS3 do not show a specific size bias but vary in their abundance levels.** Size distribution of sRNAs derived from Ah04mir1887 **(A)** and Ah20mirS3 **(B)**. **(C-D)** Genomic representation of the Ah04mir1887 and Ah20mirS3 loci, and their respective sRNAs. sRNAs are colored according to their abundance level.
(TIFF)

**S9 Fig. Molecular features of *A. halleri*-specific sRNAs produced by Ah04mir1887 and Ah20mirS3. (A-B)** Expression of Ah04mir1887 **(A)** and Ah20mirS3 **(B)** *A. halleri* sRNAs as a function of their targeting score against *SCR01.* The dot color corresponds to the size of the sRNA. The grey box highlights sRNAs predicted to induce a reduction in *SCR01* transcript abundance (score ≥ 18). *n* corresponds to the total number of unique sRNAs identified in *A. halleri*. **(C-D)** Size distribution of *A. halleri* sRNAs derived from Ah04mir1887 **(C)** and Ah20mirS3 **(D)**. **(E-F)** Genomic representation of the Ah04mir1887 **(E)** and Ah20mirS3 **(F)** loci, and the respective *A. halleri* sRNAs. sRNAs are colored according to their abundance level. **(G)** Summary of sRNA sequencing efforts in *A. thaliana* transgenic lines and *A. halleri* lines carrying Ah04mir1887 and Ah20mirS3.
(TIFF)

**S10 Fig. 5' terminal nucleotide frequencies in Ah04mir1887 (A) and Ah20mirS3 (B) sRNAs, related to their inferred AGO loading pattern.**
(TIFF)

**S11 Fig. sRNA mapping at recessive *SCR* loci.** Genome browser snapshots showing sRNA mapping across *SCR* regions in *Arabidopsis halleri* plants. sRNAs in floral buds of plants carrying different combinations of *S*-alleles were sequenced, and their genotypes are indicated on the left. In each case, the shown *SCR* locus corresponds to the recessive allele in the respective *S*-allele combination (*SCR01* in Ah04Ah01 plants and *SCR04* in Ah20Ah04 plants). No considerable sRNA accumulation is observed at these loci, with the exception of a single intronic island corresponding to a TE region. *SCR* exons are represented by black boxes and TEs by green boxes. Pileup tracks indicate sRNA read abundance, while individual sRNAs are shown below and color-coded by strand (red, forward; blue, reverse).
(TIFF)

**S12 Fig. Size distribution, 5' terminal nucleotide frequencies, and AGO enrichment in control regions of total, AGO1, and AGO4 IP sRNA libraries from *Arabidopsis thaliana* plants expressing Ah04mir1887 or Ah20mirS3.**

sRNA features in distinct genomic regions are shown, including canonical miRNAs expressed in floral buds (miR159, miR162, miR165/166, miR167 and miR160), RdDM loci (PolIV/PolV-dependent de novo RdDM loci, Group E loci in Blevins et al.) [89], TAS loci (TAS1a/b) [90], and TEs (all annotated TEs in the TAIR10 genome assembly). The sRNA size distribution, 5' terminal nucleotide composition, and AGO enrichment follow expected patterns for each region, validating the generated sRNA libraries. AGO enrichment is represented as normalized rpm. Raw sRNA read counts are provided in S6 Table.
(TIFF)

## Acknowledgments

We thank Hervé Vaucheret, Vincent Colot, Claudia Köhler, Rebecca Mosher and Nicolas Butel for helpful discussions, and Hervé Vaucheret for sharing seeds.

## Author contributions

**Conceptualization:** Rita A. Batista, Eléonore Durand, Jacinthe Azevedo-Favory, Samson Simon, Manu Dubin, Matteo Barois, Isabelle Fobis-Loisy, Thierry Lagrange, Xavier Vekemans, Vincent Castric.

**Data curation:** Rita A. Batista, Eléonore Durand, Monika Mörchen, Jacinthe Azevedo-Favory, Vincent Castric.

**Formal analysis:** Rita A. Batista, Eléonore Durand, Monika Mörchen, Jacinthe Azevedo-Favory, Samson Simon, Manu Dubin, Matteo Barois, Sylvain Legrand, Vincent Castric.

**Funding acquisition:** Rita A. Batista, Sylvain Legrand, Ute Krämer, Thierry Lagrange, Xavier Vekemans, Vincent Castric.

**Investigation:** Rita A. Batista, Eléonore Durand, Monika Mörchen, Jacinthe Azevedo-Favory, Samson Simon, Manu Dubin, Vinod Kumar, Eléanore Lacoste, Corinne Cruaud, Christelle Blassiau, Matteo Barois, Anne-Catherine Holl, Chloé Ponitzki, Nathalie Faure, William Marande, Sonia Vautrin, Jean-Marc Aury, Sylvain Legrand, Vincent Castric.

**Methodology:** Rita A. Batista, Eléonore Durand, Monika Mörchen, Jacinthe Azevedo-Favory, Samson Simon, Manu Dubin, Matteo Barois, William Marande, Vincent Castric.

**Project administration:** Vincent Castric.

**Resources:** Rita A. Batista, Jacinthe Azevedo-Favory, Sylvain Legrand, Ute Krämer, Thierry Lagrange, Xavier Vekemans, Vincent Castric.

**Software:** Rita A. Batista, Eléonore Durand, Manu Dubin, Eléanore Lacoste, Matteo Barois, Sylvain Legrand.

**Supervision:** Eléonore Durand, Jacinthe Azevedo-Favory, Christelle Blassiau, Jean-Marc Aury, Ute Krämer, Thierry Lagrange.

**Validation:** Rita A. Batista, Eléonore Durand, Monika Mörchen.

**Visualization:** Rita A. Batista, Eléonore Durand, Samson Simon, Manu Dubin, Eléanore Lacoste, Matteo Barois.

**Writing – original draft:** Rita A. Batista, Vincent Castric.

**Writing – review & editing:** Rita A. Batista, Jacinthe Azevedo-Favory, Isabelle Fobis-Loisy, Ute Krämer, Thierry Lagrange, Xavier Vekemans, Vincent Castric.

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
