## [Decision Letter · Decision Letter 0]

24 Dec 2025

PGENETICS-D-25-01243

Dominance modifiers at the Arabidopsis self-incompatibility locus retain proto-miRNA features and act through non-canonical pathways

PLOS Genetics

Dear Dr. Castric,

Thank you for submitting your manuscript to PLOS Genetics. After careful consideration, we feel that it has merit but does not fully meet PLOS Genetics's publication criteria as it currently stands. Therefore, we invite you to submit a revised version of the manuscript that addresses the points raised during the review process.

We look forward to receiving your revised manuscript.

Kind regards,

Yalong Guo, Ph.D.

Guest Editor

PLOS Genetics

Angela Hancock

Section Editor

PLOS Genetics

Aimée Dudley

Editor-in-Chief

PLOS Genetics

Anne Goriely

Editor-in-Chief

PLOS Genetics

**Journal Requirements:**

At this stage, the following Authors/Authors require contributions: Rita A. Batista, Eléonore Durand, Monika Mörchen, Jacinthe Azevedo-Favory, Samson Simon, Manu Dubin, Vinod Kumar, Eléanore Lacoste, Corinne Cruaud, Christelle Blassiau, Matteo Barois, Anne-Catherine Holl, Chloé Ponitzki, Nathalie Faure, William Marande, Sonia Vautrin, Isabelle Fobis-Loisy, Jean-Marc Aury, Sylvain Legrand, Ute Krämer, Thierry Lagrange, Xavier Vekemans, and Vincent Castric. Please ensure that the full contributions of each author are acknowledged in the "Add/Edit/Remove Authors" section of our submission form.

The list of CRediT author contributions may be found here: https://journals.plos.org/plosgenetics/s/authorship#loc-author-contributions

https://journals.plos.org/plosgenetics/s/submission-guidelines#loc-parts-of-a-submission

4) We noticed that you used the phrase 'data not shown' in the manuscript. We do not allow these references, as the PLOS data access policy requires that all data be either published with the manuscript or made available in a publicly accessible database. Please amend the supplementary material to include the referenced data or remove the references.

5) Please upload all main figures as separate Figure files in .tif or .eps format. For more information about how to convert and format your figure files please see our guidelines:

6) We have noticed that you have uploaded Supporting Information files, but you have not included a list of legends. Please add a full list of legends for your Supporting Information files after the references list.

7) Some material included in your submission may be copyrighted. According to PLOSu2019s copyright policy, authors who use figures or other material (e.g., graphics, clipart, maps) from another author or copyright holder must demonstrate or obtain permission to publish this material under the Creative Commons Attribution 4.0 International (CC BY 4.0) License used by PLOS journals. Please closely review the details of PLOSu2019s copyright requirements here: PLOS Licenses and Copyright. If you need to request permissions from a copyright holder, you may use PLOS's Copyright Content Permission form.

Potential Copyright Issues:

i) Figures 1A, and 1B. Please confirm whether you drew the images / clip-art within the figure panels by hand. If you did not draw the images, please provide (a) a link to the source of the images or icons and their license / terms of use; or (b) written permission from the copyright holder to publish the images or icons under our CC BY 4.0 license. Alternatively, you may replace the images with open source alternatives. See these open source resources you may use to replace images / clip-art:

**Reviewers' comments:**

Reviewer's Responses to Questions

**Comments to the Authors:**

Reviewer #1: This study investigates the biogenesis and mode of action of small RNAs that act as dominance modifiers in the hierarchy of S-locus alleles of Arabis. To do so, the authors have reconstituted the A. halleri self-incompatibility response in Arabidopsis thaliana by expressing a cognate pair of SRK and SCR alleles, with or without small-RNA loci that are thought to target and suppress expression of the particular SCR allele used. By combining this system with mutations in the canonical RdDM pathway, the authors show that the sRNA-mediated reduction in SCR transcript levels is independent of the RdDM pathway. They go on to perform sRNA sequencing on appropriate genotypes and find that the sRNA precursors are processed into a multitude of sRNAs of different length and sequence; are loaded into AGO1 or AGO4 and probably also into other unknown AGOs; and can target a large number of SCR alleles in A. halleri. Often sRNAs made from the same precursor target several SCR alleles, and many SCR allele is targeted by multiple different sRNAs. Based on this, the authors propose that the sRNA loci have retained features of proto-miRNAs that are processed into many different sRNA species and can trigger different effector pathways to silence SCR expression from recessive S-locus haplotypes.

This is a very interesting study that makes an important contribution to our understanding of the dominance hierarchy between S-locus alleles in the Brassicaceae and suggests a different mode of action of the sRNA modifiers compared to Brassica.

The following comments should be addressed to further strengthen the manuscript.

1. Figure 1: The difference between lines with and without the sRNA construct is pretty clear, but for completeness’ sake it would be good to add the result of statistical tests.

2. Predicted targeting of SCR01 promotor (lines 106-107): This is not entirely clear from Figure 1 S1, possibly due to the smoothing of the curves using a 20 nt sliding window. Are the authors certain that the miR target site is not in the 5' UTR?

3. Lines 113-116: Why did the others not make a new sRNA expressing line that only expresses Ah20mirS3? Can they please comment on this?

4. Lines 149-152 “And while…(Fig. 1D).”: This sentence is rather unclear and seems to give the wrong impression. Please rephrase.

5. Lines 152-158: Statistical tests would be good to show that the level of SCR01 expression is indeed higher in polIV and polV mutants without the sRNA than in wt without sRNA. Also, do the authors have another transgenic line for the SCR01*mir1887 construct that perhaps escapes the suggested transgene silencing? This would be useful to strengthen the authors’ interpretation versus the alternative that the sRNA-mediated silencing is less efficient in the polIV and polV backgrounds.

6. Line 176: Should this not be mir1887?

7. Figure 3: It is not clear to me why mir1887 is not considered in the dominance interactions by the Ah20 allele. Can the authors please explain this? Should small RNAs derived from the mir1887 locus not also be made in plants heterozygous for Ah20?

8. Lines 286-289: Is there any chance of repeating the experiment in an ago1 ago4 ago6 triple mutant background?

9. Lines 342 onwards: What is known about the evolutionary age of the sRNA loci mediating the dominance hierachy? In other words, for how long have they maintained their proto-miRNA features? Can closely related sRNA loci like mir1887 and mirS3 also be found at the genome-sequence level for example in C. grandiflora? If yes, this would strengthen the argument that their proto-miRNA features are functionally relevant. If not, the alternative would be that the ones in A. halleri are simply young and haven't been streamlined by evolution yet to produce single, well-defined sRNAs.

10. It is a real pity that DNA methylation of the transgenes has not been directly tested. I can see the authors’ argument that this should only be expected in the tapetum, and so should be difficult to study in bulk samples. I also appreciate that setting up FACS or INTACT to isolate tapetum nuclei is beyond the scope of the present study. However, doing bisulfite sequencing from young anthers at a high coverage should be able to detect methylated DNA from the subpopulation of tapetum cells. This seems quite feasible and doing so would greatly strengthen the manuscript.

Reviewer #2: Batista et al. report a molecular dissection of the sRNA-based self-incompatibility (SI) dominance across A. halleri alleles. To this end, the authors recreated the self-incompatibility reaction in the model plant A. thaliana by introducing different A. halleri alleles for the SRK01 receptor and its cognate SCR01 ligand in the C24 accession. They then introduced distinct dominant A. halleri sRNA precursor loci to test their molecular pathways as well as the incompatibility reactions. The authors further tested this system in mutant backgrounds for distinct components of canonical RNA-dependent DNA methylation pathways and found that these mutants do not alter the incompatibility reactions, excluding a role for canonical RdDM. This is an important result. The authors performed additional analyses to characterize the repertoire and AGO loading of siRNAs produced by the sRNA precursor loci, and conclude that they operate through an unconventional post-transcriptional gene silencing pathway, triggered by miRNA-like precursors.

Overall, the manuscript presents a clever design and a powerful genetic system that will be important for researchers studying the genetic basis of self-incompatibility. Most of the conclusions are well supported, and the discussion is very appropriate.

I have nonetheless identified some issues that the authors should revise.

Fig. 1C and Fig. 1D. The plots currently show differences, but the results are lacking statistical testing. Please include which test was used, what the n represents, and how multiple comparisons were handled. Importantly, it is unclear whether siliques are treated as independent experimental units or pseudo-replicates, and how many independent transgenic lines were used for each experiment.

Fig. 1D (polIV/polV backgrounds). Introduction of Ah20mirS3 in polIV and polV backgrounds still appears to yield an incompatible pollen phenotype (low pollen germination). This is unexpected if the interpretation is that the effect is independent of canonical RdDM. Please clarify the reason for this outcome (e.g., elevated basal SCR01 transcript levels in these mutants and incomplete reduction below the phenotypic threshold), and ensure that the text and figure are fully consistent.

sRNA pathway benchmarks. The authors reports atypical size/processing features for dominance modifier sRNAs, but this claim is lacking controls, For instance, does the atypical siRNAs profile is biological, or an artifact of the library? The Authors should include analyses of siRNAs size and 5′ nucleotide composition for representative canonical RdDM (24-nt heterochromatic siRNAs), PTGS/RDR6-dependent siRNAs (e.g., ta-siRNAs), and annotated miRNAs, alongside the S-locus-derived sRNAs. If AGO-IP is used to infer loading, it would also help to show expected enrichment of these “canonical targets” in AGO4 vs AGO1 as positive controls for the IPs.

The authors interpret the results as being mediated by an unconventional PTGS pathway. However, I wonder whether miRNA-like trigger followed by transitivity (secondary siRNA production and amplification on the SCR01 transcript) could also fit. Please provide additional analyses to assess whether SCR01 shows signatures of transitivity upon expression of the dominance modifiers, including (i) mapping and coverage plots of sRNAs along the SCR01 transcript and flanking regions in the presence vs absence of Ah20mirS3/Ah04mir1887, (ii) size-class and strand-bias profiles of SCR01-mapping sRNAs (in particular, whether new 21–22 nt species appear and spread away from the predicted target site), and (iii) if feasible, an analysis of phasing or positional enrichment relative to the predicted target site. These analyses would help clarify whether amplification via secondary siRNAs could contribute to the observed SCR01 downregulation.

“PTGS unlikely because the target is promoter/intron”. The argument that PTGS is unlikely because PTGS substrates are “generally thought to be the mature mRNA” is not fully convincing on its own, since nuclear/transcript-associated pathways could still operate. Indeed, the manuscript later invokes non-canonical pathways…. Please qualified the “PTGS unlikely” claim and explicitly discuss alternative RNA-based scenarios that remain compatible with the data, for example promoter-proximal transcripts and transitivity.

Many conclusions in the manuscript rely on a hard cutoff (score ≥ 18) to define “functional” sRNAs and build the interaction network. It would be interesting to provide as sensitivity test to show whether the conclusions are robust to changes in the threshold (e.g., 16, 17, 18, 19, 20...) and clarify how mismatches and G:U pairs are weighted in the scoring (even if detailed in the prior reference, a short reminder in the methods section would help).

It would be interesting to test DNA methylation of interactions A and B to rule out unconventional RdDM-directed TGS.

Reviewer #3: Review is uploaded as an attachment

**Have all data underlying the figures and results presented in the manuscript been provided?**

Large-scale datasets should be made available via a public repository as described in the *PLOS Genetics*
data availability policy, and numerical data that underlies graphs or summary statistics should be provided in spreadsheet form as supporting information., and numerical data that underlies graphs or summary statistics should be provided in spreadsheet form as supporting information., and numerical data that underlies graphs or summary statistics should be provided in spreadsheet form as supporting information., and numerical data that underlies graphs or summary statistics should be provided in spreadsheet form as supporting information.

Reviewer #1: None

Reviewer #2: Yes

Reviewer #3: Yes

PLOS authors have the option to publish the peer review history of their article (what does this mean?). If published, this will include your full peer review and any attached files.). If published, this will include your full peer review and any attached files.). If published, this will include your full peer review and any attached files.). If published, this will include your full peer review and any attached files.

...

Reviewer #1: No

Reviewer #2: No

Reviewer #3: No

**Figure resubmission:**
---

## [Decision Letter · Decision Letter 1]

9 Apr 2026

Dear Dr Castric,

We are pleased to inform you that your manuscript entitled "Dominance modifiers at the Arabidopsis self-incompatibility locus retain proto-miRNA features and act through non-canonical pathways" has been editorially accepted for publication in PLOS Genetics. Congratulations!

Yours sincerely,

Yalong Guo, Ph.D.

Guest Editor

PLOS Genetics

Angela Hancock

Section Editor

PLOS Genetics

Aimée Dudley

Editor-in-Chief

PLOS Genetics

Anne Goriely

Editor-in-Chief

PLOS Genetics

BlueSky: @plos.bsky.social

Comments from the reviewers (if applicable):

Reviewer's Responses to Questions

**Comments to the Authors:**

Reviewer #1: This is an improved manuscript that fully addresses all of my previous comments.

Reviewer #2: The authors have successfully addressed all my comments. I have no further concerns

Reviewer #3: Authors have sufficiently addressed my comments and concerns. I am particularly excited to see that the statistical analyses are consistent with original interpretations, and believe their inclusion significantly increases the overall rigor and confidence in these results.

**Have all data underlying the figures and results presented in the manuscript been provided?**

Large-scale datasets should be made available via a public repository as described in the *PLOS Genetics*
data availability policy, and numerical data that underlies graphs or summary statistics should be provided in spreadsheet form as supporting information., and numerical data that underlies graphs or summary statistics should be provided in spreadsheet form as supporting information., and numerical data that underlies graphs or summary statistics should be provided in spreadsheet form as supporting information., and numerical data that underlies graphs or summary statistics should be provided in spreadsheet form as supporting information.

Reviewer #1: Yes

Reviewer #2: Yes

Reviewer #3: Yes

PLOS authors have the option to publish the peer review history of their article (what does this mean?). If published, this will include your full peer review and any attached files.). If published, this will include your full peer review and any attached files.). If published, this will include your full peer review and any attached files.). If published, this will include your full peer review and any attached files.

...

Reviewer #1: No

Reviewer #2: No

Reviewer #3: No

**Data Deposition**

If you have submitted a Research Article or Front Matter that has associated data that are not suitable for deposition in a subject-specific public repository (such as GenBank or ArrayExpress), one way to make that data available is to deposit it in the Dryad Digital Repository. As you may recall, we ask all authors to agree to make data available; this is one way to achieve that. A full list of recommended repositories can be found on our . As you may recall, we ask all authors to agree to make data available; this is one way to achieve that. A full list of recommended repositories can be found on our . As you may recall, we ask all authors to agree to make data available; this is one way to achieve that. A full list of recommended repositories can be found on our . As you may recall, we ask all authors to agree to make data available; this is one way to achieve that. A full list of recommended repositories can be found on our website....

http://datadryad.org/submit?journalID=pgenetics&manu=PGENETICS-D-25-01243R1

Additionally, please be aware that our data availability policy requires that all numerical data underlying display items are included with the submission, and you will need to provide this before we can formally accept your manuscript, if not already present. requires that all numerical data underlying display items are included with the submission, and you will need to provide this before we can formally accept your manuscript, if not already present. requires that all numerical data underlying display items are included with the submission, and you will need to provide this before we can formally accept your manuscript, if not already present. requires that all numerical data underlying display items are included with the submission, and you will need to provide this before we can formally accept your manuscript, if not already present.

**Press Queries**

If you or your institution will be preparing press materials for this manuscript, or if you need to know your paper's publication date for media purposes, please inform the journal staff as soon as possible so that your submission can be scheduled accordingly. Your manuscript will remain under a strict press embargo until the publication date and time. This means an early version of your manuscript will not be published ahead of your final version. PLOS Genetics may also choose to issue a press release for your article. If there's anything the journal should know or you'd like more information, please get in touch via plosgenetics@plos.org....

---

## [Editor Report · Acceptance letter]

PGENETICS-D-25-01243R1

Dominance modifiers at the Arabidopsis self-incompatibility locus retain proto-miRNA features and act through non-canonical pathways

Dear Dr Castric,

We are pleased to inform you that your manuscript entitled "Dominance modifiers at the Arabidopsis self-incompatibility locus retain proto-miRNA features and act through non-canonical pathways" has been formally accepted for publication in PLOS Genetics! Your manuscript is now with our production department and you will be notified of the publication date in due course.

With kind regards,

Anita Estes

PLOS Genetics

On behalf of:
